# SARS-CoV-2-mediated dysregulation of metabolism and autophagy uncovers host-targeting antivirals

Viruses manipulate cellular metabolism and macromolecule recycling processes like autophagy. Dysregulated metabolism might lead to excessive inflammatory and autoimmune responses as observed in severe and long COVID-19 patients. Here we show that SARS-CoV-2 modulates cellular metabolism and reduces autophagy. Accordingly, compound-driven induction of autophagy limits SARS-CoV-2 propagation. In detail, SARS-CoV-2-infected cells show accumulation of key metabolites, activation of autophagy inhibitors (AKT1, SKP2) and reduction of proteins responsible for autophagy initiation (AMPK, TSC2, ULK1), membrane nucleation, and phagophore formation (BECN1, VPS34, ATG14), as well as autophagosome-lysosome fusion (BECN1, ATG14 oligomers). Consequently, phagophore-incorporated autophagy markers LC3B-II and P62 accumulate, which we confirm in a hamster model and lung samples of COVID-19 patients. Single-nucleus and single-cell sequencing of patient-derived lung and mucosal samples show differential transcriptional regulation of autophagy and immune genes depending on cell type, disease duration, and SARS-CoV-2 replication levels. Targeting of autophagic pathways by exogenous administration of the polyamines spermidine and spermine, the selective AKT1 inhibitor MK-2206, and the BECN1-stabilizing anthelmintic drug niclosamide inhibit SARS-CoV-2 propagation in vitro with $IC_{50}$ values of 136.7, 7.67, 0.11, and 0.13 µM, respectively. Autophagy-inducing compounds reduce SARS-CoV-2 propagation in primary human lung cells and intestinal organoids emphasizing their potential as treatment options against COVID-19.

Severe acute respiratory syndrome coronavirus 2 (SARS-CoV-2) poses an imminent threat to global health. As of 18 May 2021, >162,000,000 individuals were infected in >220 countries, with >3,375,000 fatalities[1]. SARS-CoV-2 infections cause CoV infectious disease 19 (COVID-19) in humans, which is associated with common cold symptoms (dry cough, fever, fatigue, shortness of breath, sore throat), but also with olfactory and taste disorders, dyspnoea, and severe pneumonia[2].

Compound-based targeting of cellular proteins that are essential for the virus life cycle has led to the discovery of broadly reactive drugs against a range of CoV[3]. However, undirected approaches often provide limited insight into the molecular and functional details on how the drugs affect virus propagation[4,5]. As virus propagation depends on energy and metabolic substrates of host cells, drug target identification should consider the metabolism of infected cells[3]. Dysregulations and age-dependent decline of multiple metabolic processes in humans are associated with comorbidities like diabetes, hypertension, and obesity, which are believed to influence COVID-19 severity[6]. Catabolic recycling processes such as the ubiquitin-proteasome system (UPS) and macro-autophagy (hereafter referred to as autophagy) maintain energy and protein homeostasis in cells[7]. The UPS facilitates ubiquitin-targeted rapid degradation of proteins via the proteasome[8]. Autophagy, which is tightly controlled by metabolism, is a highly conserved lysosomal degradation process of long-lived proteins, lipids, and organelles in eukaryotic cells[9,10]. The mechanistic target of rapamycin complex 1 (mTORC1) and AMP-activated protein kinase (AMPK) regulate autophagy by responding to metabolic alterations via continuous crosstalk with glucose and protein homeostasis[11]. mTORC1 activity is, for example, increased in response to high intracellular amino acid levels, stimulates anabolic cell growth, and inhibits autophagy to reduce untargeted protein degradation[12]. Increased AMP and low glucose levels trigger AMPK, indicating energy and amino acid deficiency, and resulting in autophagy activation[13]. Mitochondrial turnover (mitophagy) and reactive oxygen species (ROS)-dependent polyamine pathways also promote autophagy[14,15]. Polyamines are generated as part of a gradual amino acid degradation process turning arginine to ornithine and subsequently via putrescine to spermine and spermidine. Both, spermine and spermidine, can be converted to putrescine via their acetylated forms (NAcspm, NAcspd) by spermidine-spermine acetyltransferase (SAT1), promoting cellular export of polyamines, and restricting replication of several RNA viruses[16–18]. Spermidine hypusinates eukaryotic translation initiation factor eIF5A[15], which functions globally in translation elongation and termination and activates lysosomal biogenesis as well as autophagy transcription factor EB (TFEB). TFEB is a key regulator for proteins responsible for autophagy and mitochondrial respiration[15,19].

During autophagy, intracellular macromolecules are recycled by incorporation into microtubule-associated proteins 1 A/1B light chain 3B (LC3B)-lipidated autophagosomes (AP) and degradation into their monomers, such as fatty and amino acids, after fusion with acidic lysosomes[20]. In the case of highly pathogenic Middle East respiratory syndrome (MERS)-CoV, we recently showed that autophagy is limited by a virus-induced AKT1-dependent activation of S-phase kinase-associated protein 2 (SKP2), an E3-ligase, which targets the key autophagy-initiating protein Beclin-1 (BECN1) for proteasomal degradation[21]. BECN1 degradation led to reduced ATG14-dependent fusion of AP with lysosomes[22] and resulted in accumulated AP, lipidated LC3B (LC3B-II), and P62 levels through blockage of autophagic flux. We assumed that MERS-CoV-induced reduction of autophagy might prevent degradation of virus proteins and enhance CoV-driven exploitation of lipid resources required for the production of infectious virus particles[23,24]. Finally, inhibition of SKP2 by different compounds, including clinically approved drugs, stabilized BECN1

and limited MERS-CoV propagation[21], indicating that autophagy-inducing compounds hold promise as antiviral drugs.

However, detailed mechanisms on how highly pathogenic CoV accomplish metabolic reprogramming and secure lipid and amino acid availability without activating antiviral counteractions are largely unknown. In this interdisciplinary and translational study we investigate the impact of SARS-CoV-2 infection on cell metabolism and the downstream effects on catabolic autophagy in vitro, ex vivo, and in vivo, thereby identifying multiple targets for the application of approved drugs and the development of antiviral therapies.

## Results

**SARS-CoV-2 causes accumulation of key metabolites in infected cells**. To explore the putative impact of SARS-CoV-2 infection on central carbon metabolism, we performed a targeted metabolomic analysis on SARS-CoV-2-susceptible, interferon (IFN)-deficient, VeroFM monkey kidney cells and IFN-competent human lung Calu-3 cells[5]. The initial metabolite profiles were analyzed by multivariate principal component analysis (PCA), an unsupervised statistical method suitable for analyzing and classifying metabolomics datasets[25]. The profiles separated SARS-CoV-2 and control group (CoV-2 vs mock; Supplementary Fig. 1 (amine-containing) and Supplementary Fig. 2 (anion-containing)). In detail, 37/90 (VeroFM) and 55/93 (Calu-3) analyzed metabolites were significantly altered (cutoff: FDR-corrected $p < 0.05$, normalized to cell counts) upon SARS-CoV-2 infection (Supplementary Tables 1 (VeroFM), 2 (Calu-3); and Supplementary Fig. 3). Whereas in the VeroFM cells all differential metabolites were significantly increased upon SARS-CoV-2 infection, the Calu-3 cells showed 47 metabolites that were significantly increased, while 8 metabolites were significantly decreased (Supplementary Fig. 3, Supplementary Tables 1–2). The differentially regulated metabolites comprised 15 metabolic pathways (Fig. 1a, VeroFM; Fig. 1b, Calu-3). Major changes were linked to amino acid pool sizes, comprising phenylalanine (phe), tyrosine (tyr), glycine (gly), serine (ser), and threonine (thr) metabolism, but also to metabolites of glyoxylate, dicarboxylate, pantothenate, and pyrimidine metabolism. High systemic amino acid levels (Fig. 1c, orange) might be explained by different mechanisms including SARS-CoV-2 NSP1-mediated translational shutdown[26,27], modulated catabolic processes, or increased amino acid biosynthesis. To explore if high amino acid levels result from de novo biosynthesis, we performed U-$^{13}C_6$ glucose isotope enrichment analysis in SARS-CoV-2- and mock-infected Calu-3 cells (Supplementary Fig. 4a, b; Supplementary Tables 3–4). As relative $^{13}C$ enrichment of the most abundant amino acids and intermediates of glycolysis and the TCA cycle (Supplementary Fig. 4a, b) showed minor differences between the SARS-CoV-2- and mock-infected cells, we focused our analyses on catabolic protein degradation via the UPS or autophagy. SARS-CoV-2-induced catabolic UPS, as shown by a global increase of K48-ubiquitinated proteins (Supplementary Fig. 5a), likely increases the availability of proteasome-derived amino acids necessary for virus production[28]. Congruently, inhibition of proteasomal degradation by MG132 reduced infectious virus particle production (Supplementary Fig. 5b). As shown for MERS-CoV, limited autophagy might additionally reduce the overall rate of host protein degradation which is supported by the high levels of amino acids and nucleoside triphosphates (Fig. 1c, orange and blue; ATP, GTP, CTP, UTP). Furthermore, high levels of putrescine (Fig. 1c, red) hint towards increased SAT1 activity, which is a known autophagy inducer[15]. Re-analyzed bulk and single-cell RNA-sequencing datasets of Calu-3 cells[29] show increased *SAT1* expression in SARS-CoV-2-infected cells (Supplementary Fig. 5c). *SAT1* is a known IFN-stimulated gene[18] and its upregulation was

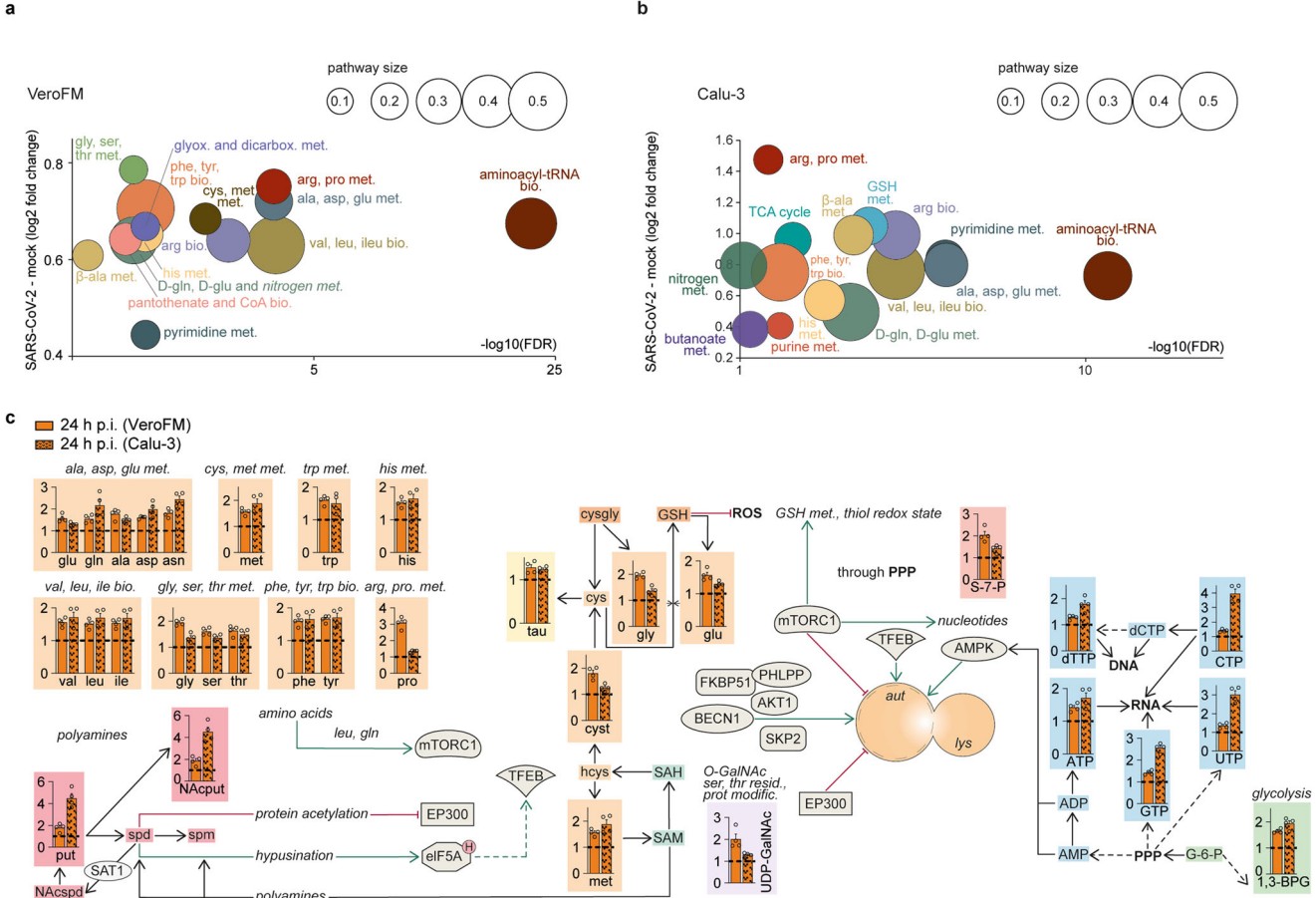

**Fig. 1 SARS-CoV-2 causes accumulation of key metabolites in infected cells. a, b** Analysis and regulation of significantly altered pathways of mock- and SARS-CoV-2-infected (24 h p.i., MOI = 0.1) VeroFM cells (**a**) or Calu-3 cells (**b**). The y-axis shows the (median) log2 fold change (FC) of all significantly altered metabolites of the indicated pathway while the –log10 corrected p-value (false discovery rate (FDR)) is shown on the x-axis. The size of the circles illustrates the number of significantly changed metabolites in relation to all metabolites of a specific pathway. **c** Analysis of the autophagic pathway and the involved metabolites: 'amino acids' and 'GSH metabolism' (orange), 'nucleotides' (blue), 'glycolysis' (green), 'polyamine metabolism' (red) and 'O-GalNAcylation' (purple) in mock- and SARS-CoV-2-infected (24 h p.i.) VeroFM and Calu-3 cells. For **a–c** Error bars represent SEM. n = 4 biological samples per group of one experiment. All p-values were determined by a two-way ANOVA and Tukey´s post hoc test. FDRs were adjusted using the Benjamini-Hochberg method. Abbreviations: 1,3-BPG, 1,3-bisphosphoglyceric acid; 3-PGA, 3-phosphoglyceric acid; ADP, adenosine diphosphate; AKT1, RAC-alpha serine/threonine-protein kinase; ala, alanine; AMP, adenosine monophosphate; AMPK, AMP-activated protein kinase; arg, arginine; asn, asparagine; asp, asparagine; ATP, adenosine triphosphate; aut, autophagosome; BECN1, beclin-1; bio., biosynthesis; CTP, cytidine triphosphate; CoA, coenzyme A; cys, cysteine; cysgly, cysteinylglycine; cyst, cystathionine; dCTP, deoxycytidine triphosphate; dicarbox., dicarboxylate; eIF5AH, eukaryotic translation initiation factor 5A hypusinated; EP300, histone acetyltransferase p300; F1P, fructose 1-phosphate; F6P, fructose 6-phosphate; F-1,6-BP, fructose 1,6-bisphosphate; FKBP51, 51 kDa FK506-binding protein; G6-P, glucose 6-phosphate; gln, glutamine; glu, glutamic acid; gly, glycine; glyox., glyoxylate; GSH, glutathione (reduced); GTP, guanosine triphosphate; hcys, homocysteine; his, histidine; ile, isoleucine; lac, lactic acid; leu, leucine; lys, lysine; mal, malic acid; met, methionine; met., metabolism; modific., modification; mTORC1, mechanistic target of rapamycin complex 1; NAcput, N-acetylputrescine; NAcspd, N-acetylspermidine; orn, ornithine; PEP, phosphoenolpyruvic acid; phe, phenylalanine; PHLPP, PH domain leucine-rich repeat-containing protein phosphatase; PPP, pentose phosphate pathway; pro, proline; prot., protein; put, putrescine; pyr, pyruvic acid; resid., residue; ROS, reactive oxygen species; S7P, sedoheptulose-7-phosphate; SAH, S-adenosylhomocysteine; SAM, S-adenosylmethionine; SAT1, diamine acetyltransferase 1; SKP2, S-phase kinase-associated protein 2; ser, serine; spd, spermidine; spm, spermine; tau, taurine; TFEB, transcription factor EB; thr, threonine; trp, tryptophan; tyr, tyrosine; UDP-GalNAc, UDP-N-acetylgalactosamine; UDP, uridine diphosphate; UTP, uridine triphosphate; val, valine.

comparable to other prototypic innate immune genes (*IFIT1*, *MX1*, *MX2*; Supplementary Fig. 5d) in SARS-CoV-2-infected cells, which confirms the proposed connection between the innate immune response and the polyamine metabolism. A SAT1-induced shift of spermidine to putrescine might limit hypusination of eIF5A and downstream activation of autophagy genes[15]. Indeed, transcriptome analysis showed that the majority (n = 399) of the 471 TFEB-regulated autophagy target genes from the CLEAR network[30] were downregulated in SARS-CoV-2-infected Calu-3 cells (Supplementary Fig. 5e). Interestingly, *MAP1LC3B* and *SQSTM1* encoding key autophagy markers LC3B and P62

remained unaffected whereas *CD63* and *RAB7A*, which are both associated with vesicular and membrane trafficking[30] were down-regulated (Supplementary Fig. 5f).

**SARS-CoV-2 limits autophagy signaling and blocks autophagic flux.** In our metabolomics dataset we identified key metabolites, which regulate the activity of autophagy-modulating AMPK (AMP/ATP ratio) and mTORC1 (amino acids). Next, we aimed to investigate SARS-CoV-2-specific effects on autophagy-linked proteins by applying previously established assays[21] according to the expert-curated guidelines for monitoring autophagy[31]. We,

therefore, monitored protein expression levels by Western blot analyses after infection of VeroFM cells with SARS-CoV-2 or heat-inactivated virus as control (Fig. 2a). Based on our previous experience with MERS-CoV[21], we chose a low MOI of 0.0005 and time points 8, 24, and 48 h post infection (h p.i.) to dissect autophagy during the exponential growth of SARS-CoV-2 (Supplementary Fig. 6a). SARS-CoV-2 infection influenced most of the analyzed components of the AMPK/mTORC1 pathway (Fig. 2a). Phosphorylated, active forms of AMPK (pAMPK$^{T172}$), AMPK substrates (LXRXX(pS/pT)), and AMPK downstream targets (pTSC2$^{S1387}$ and pULK1$^{S556}$) were reduced upon SARS-CoV-2 infection, suggesting that a high ATP/AMP ratio is maintained via a virus-induced decrease of host protein translation. The metabolomics data (see Fig. 1) also revealed cellular amino acid sufficiency, consistent with high levels of mTORC1 activity. Indeed, mTORC1-dependent phosphorylation of ULK1 at S757 was maintained around the basal level throughout the course of infection (Fig. 2a, ULK1$^{S757}$), suggesting that SARS-CoV-2 avoids triggering starvation-induced autophagy. In addition, we found increased levels of pAKT1$^{S473}$, which may be explained by SARS-CoV-2-induced growth factor signaling in line with previously described elevated class I phosphatidylinositol 3-kinase (PI3K) activity and AKT1 phosphorylation[32]. AKT1 activates the BECN1-targeting E3-ligase SKP2 through phosphorylation at S72[21]. Congruently, CDK2-dependent and SKP2-activating phosphorylation of SKP2 at position S64[33] was enhanced (Fig. 2a, pSKP2$^{S64}$). Consequently, we found low BECN1 levels and reduced ULK1-dependent pBECN1$^{S15}$, which is in agreement with subsequently reduced ATG14 phosphorylation (pATG14$^{S29}$) and ATG14 homo-oligomerization (Fig. 2a). BECN1 is a key factor of the phagophore-nucleation complex consisting of VPS34, ATG14, AMBRA1, and p115[34]. In line with BECN1 reduction, we also detected reduced GFP-FYVE signals, an established reporter for VPS34 (class III PI3K) activity[35], in SARS-CoV-2-infected VeroFM cells (Supplementary Fig. 6b). The above-mentioned reduction of ATG14 oligomers, known to facilitate fusion via SNARE proteins SNAP29 and STX17[21], and promoting fusion of APs with lysosomes[22], indicated a disruption of vesicular fusion and reduction of autophagic flux. By using a well-established autophagy reporter assay, we assessed autophagosome-lysosome (AL) fusion, a hallmark of autophagic flux[36]. The number of low-pH ALs (in red) in mRFP and pH-sensitive EGFP dual-tagged LC3B-expressing and SARS-CoV-2-infected VeroFM (Fig. 2b, Supplementary Fig. 6c) and human bronchiolar NCI-H1299 cells (Supplementary Fig. 6d) was reduced compared to mock-infected cells indicating a virus-induced reduction of AL fusion. In VeroFM cells, the total number of ALs was reduced from a mean of 58 to 39 vesicles per cell, whereas the number of APs per cell was comparable (mock = 25 vesicles/cell; SARS-CoV-2 = 27 vesicles/cell). The ratio of APs to ALs shifted in SARS-CoV-2-infected cells, indicating relative AP accumulation compared to AL (mock = 0.431 ± 0.050; SARS-CoV-2 = 0.674 ± 0.076) (Fig. 2b, Supplementary Fig. 6c-d). The reduction of autophagic flux was confirmed by elevated levels of the autophagy receptor P62 in SARS-CoV-2-infected cells (Fig. 2c; Supplementary Figure 7a, NCI-H1299). As transcriptional regulation of P62-mRNA was minor in SARS-CoV-2-infected VeroFM cells at 8 and 24 h p.i. and max. 3-fold at 48 h p.i. (Supplementary Fig. 7b), the lack of AL fusion was most likely responsible for increased P62 protein levels. For further confirmation and to assess the effect of the proposed SARS-CoV-2-induced block of the autophagic flux, we co-treated SARS-CoV-2-infected VeroFM and NCI-H1299 cells with optimized amounts of bafilomycin A1 (BafA1, 100 nM, Supplementary Fig. 7c) and determined LC3B-II levels as a marker for phagophore/AP accumulation (Fig. 2d, VeroFM and Supplementary Fig. 7d, NCI-H1299). BafA1 is an inhibitor of vacuolar

H$^+$-ATPase (vATPase), an enzyme essential for lysosome acidification and degradation of AP cargo. BafA1 treatment prevents the formation of autophagy-active ALs from AL fusion[31]. Immunoblot analysis of LC3B-II in the presence of 100 nM BafA1 indicated that mock-infected cells showed a significant increase of the LC3B-II/actin ratio due to the BafA1-induced inhibition of autophagic flux, whereas SARS-CoV-2 infection alone was sufficient to elevate the LC3B-II/actin ratio (Fig. 2d, Supplementary Fig. 7d, NCI-H1299). At 8 h p.i., LC3B-II levels were slightly elevated in SARS-CoV-2-infected cells, but could still be enhanced by BafA1 treatment. At 24 and 48 h p.i., vehicle- or BafA1-treated and SARS-CoV-2-infected cells showed equally limited autophagic flux (Fig. 2d). In sum, SARS-CoV-2 infection activated negative regulators of autophagy, reduced autophagy-enhancing proteins, and limited autophagic flux by diminishing BECN1/ATG14-dependent AL fusion.

## SARS-CoV-2 replication-dependent and cell-specific regulation of autophagy in vivo

To explore if SARS-CoV-2 reduces autophagic flux in vivo, we analyzed lung samples from SARS-CoV-2-infected Syrian hamsters[37]. Protein samples were taken at 2, 3, 5, and 14 days post infection. As described previously, virus propagation in the lung reached the highest titers at days 2 and 3 (10$^8$ PFU/g) with subsequent reduction to 10$^5$ PFU/g at day 5 followed by viral clearance at days 7 and 14[37]. Particularly in aged hamsters, we found highly increased P62 and LC3B-II protein levels at day 3 (Fig. 3a). At the same time, mRNA levels of Map1lc3b and Sqstm1 remained overall unchanged throughout the course of infection (Supplementary Fig. 8) suggesting that increased P62 and LC3B-II levels were not caused by transcriptional upregulation, but rather accumulated by a SARS-CoV-2-induced blockage of autophagy. Furthermore, we compared postmortem lung samples of COVID-19 patients ($n = 6$) with lung sections of other pneumonia patients ($n = 3$) or deceased due to other causes ($n = 3$) by immunohistochemical staining of P62 and LC3B (Supplementary Tables 5, 6). More LC3B- and particularly P62-positive cells were identified in COVID-19 patient lung samples, which might be explained by increased MAP1LC3B and SQSTM1 mRNA levels or again by a reduced autophagic flux and protein accumulation (Fig. 3b, Supplementary Table 6). To distinguish between these possible explanations, we performed sNuc-seq-based transcriptional analyses for seven deceased individuals with severe COVID-19, of which three deceased early (<14 days) and four late (>14 days) post onset of symptoms (Supplementary Table 7). Cell types were defined as previously described[38] (Supplementary Fig. 9a–i). The analyses focused on autophagy-affected genes (SQSTM1, MAP1LC3B, CD63, RAB7A), ROS-induced NQO1 to monitor NRF-2 activation, autophagy-affecting polyamine-regulating SAT1 and IFN-stimulated genes (ISGs) MX1, IFIT1, ISG15 and IFITM3 in multiple cell types (Supplementary Fig. 9j) especially secretory, ciliated and AT2 cells, as the latter two are the predominant target cells for SARS-CoV-2 infection[39] (Fig. 3c). In secretory cells (Fig. 3c, left), most target genes were comparably expressed between control and early-deceased COVID-19 patients but downregulated in cells from late-deceased individuals. In the main SARS-CoV-2 target cells (ciliated, AT2), we observed a similar pattern between early- and late-deceased patients. However, mRNA levels for IFN-stimulated and autophagy genes were much more increased compared to secretory cells and were highly comparable to a cellular subpopulation in which high viral RNA levels were detected (Fig. 3c, SARS2high, Supplementary Fig. 9d). SAT1 mRNA levels were similar to those of all other ISGs, further supporting a link between IFN response and polyamine-dependent autophagy interventions (see Fig. 1c). As early- compared to late-deceased patients showed higher virus RNA levels (>10$^5$ virus RNA copies per 10,000

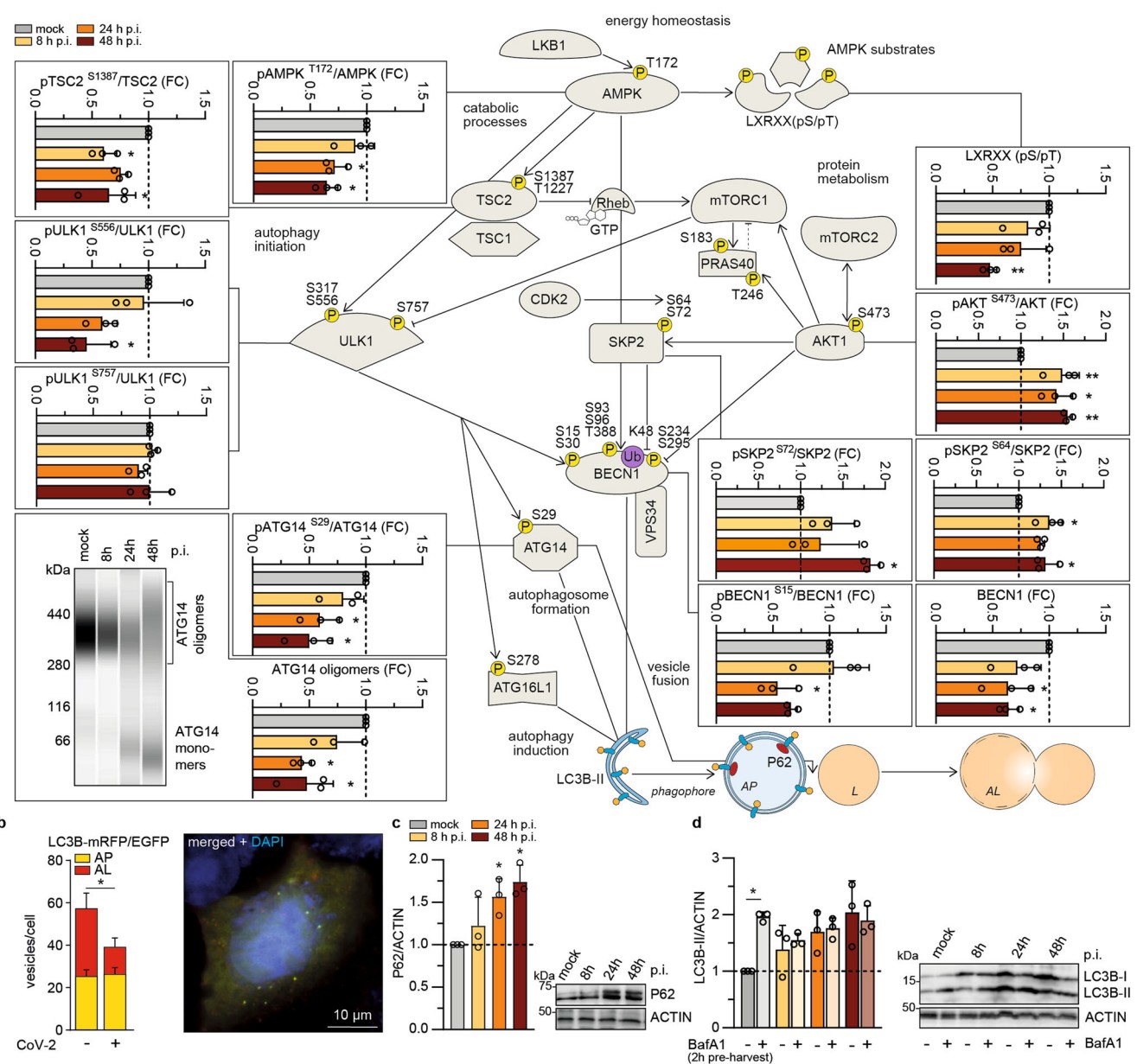

**Fig. 2 SARS-CoV-2 limits autophagy signaling and blocks autophagic flux. a** Protein levels and phosphorylation status of selected autophagy-relevant proteins in SARS-CoV-2-infected VeroFM cells at 8 h, 24 h, or 48 h post infection (h p.i.) were analyzed by Western blotting. For analysis of ATG14 oligomers (virtual blot, bottom panel, left) proteins were cross-linked 2 h prior to cell harvest and analyzed by Wes (ProteinSimple) capillary electrophoresis 8-48 h p.i. *P*-values were determined by one-way ANOVA, Bonferroni post hoc test. **b** Fluorescence microscopy of transfected and SARS-CoV-2-infected VeroFM cells expressing pH-sensitive tandem fluorescent-tagged *LC3B-mRFP/EGFP* showed that low pH autophagolysosomes (AL, red) were reduced compared to autophagosomes (AP, green + red = yellow) in virus-infected cells. Microscopic read-out was done by a scientist blind to the experimental conditions. For mock (*n* = 44 cells) and SARS-CoV-2-infected (*n* = 46 cells) VeroFM cells were analyzed. P-values were determined by two-way ANOVA, Tukey´s post hoc test, mean with SEM. Scale bar = 10 μM. **c** Accumulation of autophagy marker P62 in SARS-CoV-2-infected VeroFM cells. **d** Elevated LC3B-II levels in SARS-CoV-2-infected and bafilomycin A1 (BafA1)-pretreated VeroFM cells indicate virus-induced autophagic flux inhibition at 8 and 24 h p.i. In all panels, error bars denote SEM derived from *n* = 3 biologically independent samples derived from one experiment. P-values were determined by two-way ANOVA, Sidak post hoc test. p ≤ 0.05 (*), p ≤ 0.01 (**), p ≤ 0.001 (***), p ≤ 0.0001 (****), p > 0.05 (not significant, ns). Abbreviations: AKT1, RAC-alpha serine/threonine-protein kinase; AMPK, AMP-activated protein kinase; ATG14, autophagy-related 14; ATG16L1, autophagy-related 16 like 1; BafA1, bafilomycin A1; BECN1, Beclin-1 protein; CDK2, cyclin-dependent kinase 2; EGFP, enhanced green fluorescent protein; LC3B, microtubule-associated protein 1 A/1B light chain 3B; LXRXX(pS/pT), phospho-AMPK substrate motif; mRFP, monomeric red fluorescent protein; mTORC1/2, mechanistic target of rapamycin complex 1/2; PRAS40, proline-rich AKT1 substrate 1; Rheb, Ras homolog enriched in brain; SKP2, S-phase kinase-associated protein 2; TSC1/2, tuberous sclerosis 1/2; ULK1, Unc-51-like kinase 1; VPS34, phosphatidylinositol 3-kinase catalytic subunit type 3.

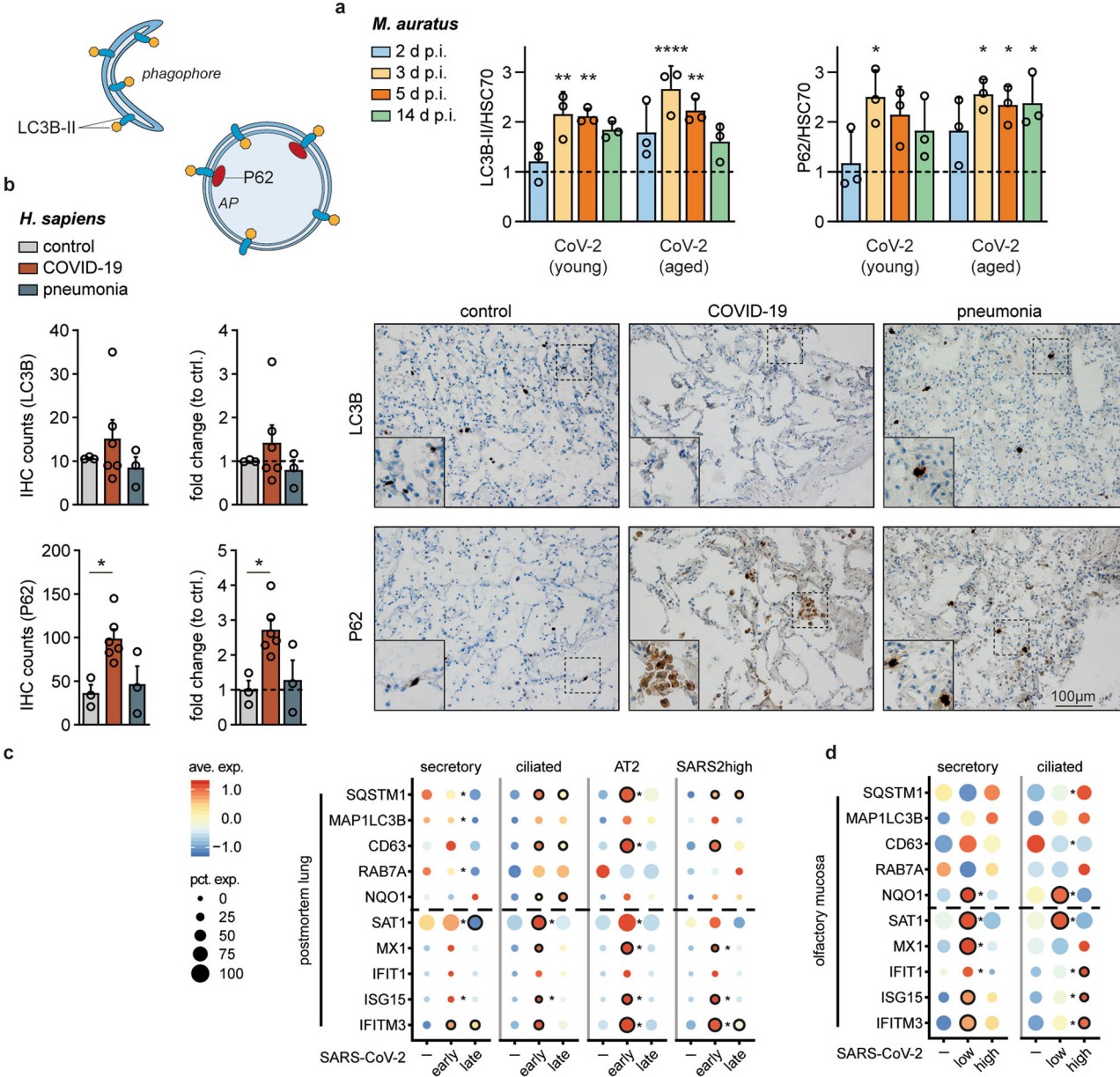

**Fig. 3 SARS-CoV-2 replication-dependent and cell-specific regulation of autophagy in vivo. a** Enhanced relative LC3B-II/HSC70 and P62 protein levels in lung samples of SARS-CoV-2-infected compared to mock-infected hamsters (dotted line). Syrian hamsters (*Mesocricetus auratus*, $n = 3$ per group, one experiment) were infected or mock-infected (heat-inactivated virus) with 1x10e5 plaque-forming units SARS-CoV-2 by intranasal instillation. Lung samples were taken at days 2, 3, 5, 14 post infection, and protein lysates were analyzed by Western blotting. P-values were determined by two-way ANOVA, Tukey´s post hoc test, mean with SD. **b** Immunohistochemistry (IHC). Increased amount of P62 and LC3B-positive cells in formalin-fixed and antibody-stained lung samples from deceased COVID-19 ($n = 6$), pneumonia ($n = 3$), and control patients ($n = 3$). For each sample, 4 randomly chosen microscopic fields (40x) were independently rated by two persons. P-values were determined by one-way ANOVA, Dunnett´s, mean with SEM. Scale bar = 100 µM. **c** Dot plots depicting scaled average expression of autophagy and immune genes for selected cell types from postmortem lung patient samples separated according to disease duration representing patients deceased within 14 days (early, $n = 3$) and after 14 days (late, $n = 4$). **d** Expression profile of distinct cell types from the olfactory mucosa of COVID-19 patients ($n = 8$) separated according to viral loads and compared to healthy controls ($n = 5$). Cut-off for the categories "low" ($n = 5$) and "high" ($n = 3$) was 10e5 GE per ml swab-derived liquid. Scaled expression levels are color-coded and the percentage of cells expressing the gene is size-coded. Differential expression analyses were calculated with MAST, and p-values were adjusted with the Benjamini-Hochberg method. For comparisons to the uninfected control, significance is indicated by a black circle. For comparisons between conditions (early vs. late or low viral load vs. high viral load) significance is depicted by an asterisk. $p \leq 0.05$ (*), $p \leq 0.01$ (**), $p \leq 0.001$ (***), $p \leq 0.0001$ (****), $p > 0.05$ (not significant, ns). Ave. Exp. = average expression, Pct. Exp. = percent of cells expressing the gene.

cells, Supplementary Table 7), we suspected that SARS-CoV-2 replication levels might influence transcriptional regulation. To further explore this, we re-analyzed single-cell sequencing (scSeq) data sets from mucosal brushes of hospitalized COVID-19 patients[38] experiencing low to moderate ($<10^5$ GE/ml, $n = 5$) or

high ($>10^5$ GE/ml, $n = 3$) virus RNA concentrations (Supplementary Table 8). Seven out of 8 patients were severely ill and all samples were taken within the main phase of virus replication (first 11 days post onset of symptoms)[40]. In patients with low SARS-CoV-2 RNA levels, we found that prototypic autophagy-related

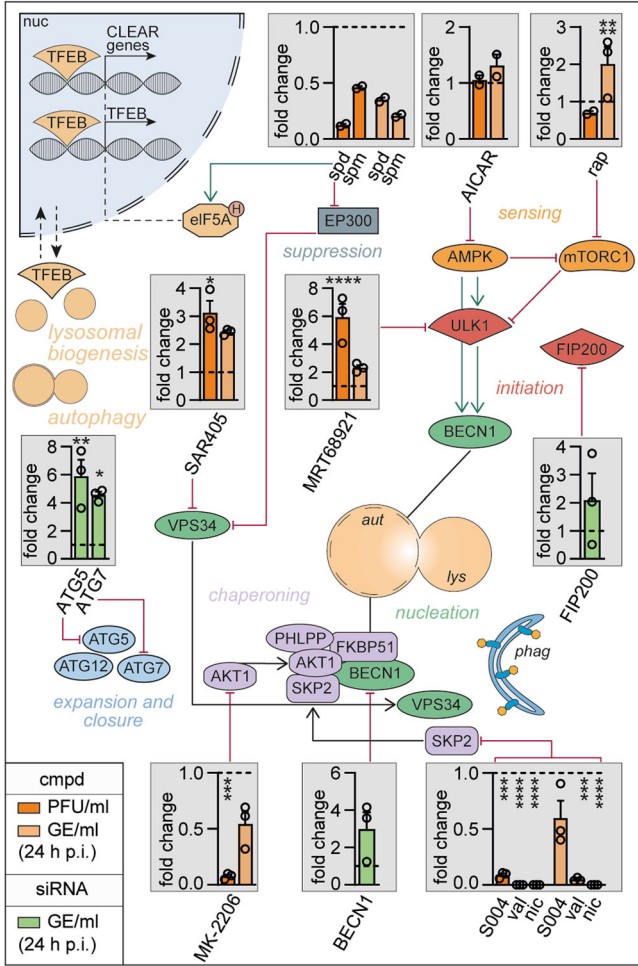

**Fig. 4 Autophagy induction limits SARS-CoV-2 growth in vitro.** Schematic representation of autophagy signaling indicating site of action of small-molecule compounds (orange columns) or siRNA knockdown (green columns) used for pathway modulation. VeroFM cells were infected with SARS-CoV-2 (MOI = 0.0005) and treated with the indicated compounds: polyamines spermidine (spd, 100 µM) and spermine (spm, 100 µM), AMPK modulator AICAR (25 µM), mTORC1 inhibitor rapamycin (rap, 300 nM), VPS34 and ULK1 inhibitors SAR405 (1 µM) and MRT68921 (5 µM), AKT1 inhibitor MK-2206 (1 µM), BECN1-stabilizers SMIP004 (S004, 10 µM), valinomycin (val, 5 µM), niclosamide (nic, 10 µM) or DMSO (vehicle, dashed lines). For siRNA knockdown, VeroFM cells were transfected with 60 nM siRNA targeting *ATG5, ATG7, FIP200, BECN1*, and infected with SARS-CoV-2 48 h later. SARS-CoV-2 infectious virus particle units (PFU) and genome equivalents (GE) per ml were determined by plaque assay and real-time RT-PCR at 24 h p.i., respectively. Data are presented as fold differences (see Supplementary Fig. 11 for unprocessed data). In all panels, error bars =SEM derived from n = 3 (n = 2 for spd, spm, AICAR) biologically independent samples from one experiment. Statistics were done for experiments with n = 3 using a two-way ANOVA, Dunnett's, and one-way ANOVA for siRNA knockdown experiments (green). Compounds with significant SARS-CoV-2 inhibition (spm, spd, MK-2206, S004, val, nic) in VeroFM cells were confirmed in Calu-3 cells (see Supplementary Fig. 11f–g). $p \leq 0.05$ (*), $p \leq 0.01$ (**), $p \leq 0.001$ (***), $p \leq 0.0001$ (****), $p > 0.05$ (not significant, ns). Abbreviations: AICAR, 5-Aminoimidazole-4-carboxamide ribonucleotide; AKT1, RAC-alpha serine/threonine-protein kinase; AMPK, AMP-activated protein kinase; ATG5, autophagy-related 5; ATG7, ubiquitin-like modifier-activating enzyme ATG7; ATG12, ubiquitin-like protein ATG12; aut, autophagosome; BECN1, Beclin-1 protein; CLEAR, coordinated lysosomal expression and regulation; cmpd, compound; EP300, histone acetyltransferase p300; eIF5AH, eukaryotic translation initiation factor 5A, hypusinated; FIP200, FAK family kinase-interacting protein of 200 kDa; FKBP51, 51 kDa FK506-binding protein; GE, genome equivalents; lys, lysosome; mTORC1, mechanistic target of rapamycin complex 1; nic, niclosamide; PFU, plaque-forming units; phag, phagophore; PHLPP, PH domain leucine-rich repeat-containing protein phosphatase; rap, rapamycin; S004, SMIP-004; SKP2, S-phase kinase-associated protein 2; spd, spermidine; spm, spermine; TFEB, transcription factor EB; ULK1, unc-51-like kinase 1; val, valinomycin; VPS34, phosphatidylinositol 3-kinase catalytic subunit type 3.

transcription of *SQSTM1, MAP1LC3B, CD63*, and *RAB7A* was upregulated in secretory cells and comparable or even reduced in putatively SARS-CoV-2-infected ciliated cells (Fig. 3d). Congruently, and similar to our in vitro metabolomics data (see Fig. 1), polyamine-regulating *SAT1* was significantly increased in ciliated cells of patients with low SARS-CoV-2 RNA concentrations (Fig. 3d), possibly facilitating downregulation of TFEB-regulated genes like *CD63*. Interestingly, in ciliated cells of patients with high SARS-CoV-2 RNA levels, ISGs were upregulated, whereas *SAT1* showed low expression levels and autophagy genes were comparably expressed or moderately upregulated, the latter being equivalent to early-deceased COVID-19 patients (see Fig. 3c). The differences in transcriptional regulations between secretory and ciliated, early- and late-deceased, as well as low and high SARS-CoV-2 RNA concentrations, imply that autophagy gene regulation is differentially controlled, depending on cell type, disease duration, and SARS-CoV-2 replication levels.

**Autophagy induction limits SARS-CoV-2 growth.** As SARS-CoV-2 modulated several metabolic pathways (see Fig. 1) and limited autophagy in vitro and in vivo (see Figs. 2 and 3), we targeted key components of the catabolic/autophagy signaling by exogenous administration of selective inhibitors, approved drugs, or cellular metabolites and explored the effect on SARS-CoV-2 propagation (Fig. 4). To exclude toxicity of the applied compounds, multiple cell viability tests (MTT, LDH, and PrestoBlue) were performed (Supplementary Fig. 10). VeroFM cells were infected and treated with different concentrations of each compound according to previous publications and cytotoxicity profile

(Supplementary Fig. 10; Supplementary Table 9). Virus growth was monitored by plaque assays to quantify the infectious particles and real-time RT-PCR to detect genome equivalents per ml (GE/ml) in cell culture supernatants 24 and 48 h p.i. (Supplementary Fig. 11a). For key autophagy proteins, siRNA-based knockdown experiments were additionally performed (Fig. 4, green columns, Supplementary Fig. 11b). The effects of all substances were monitored during the exponential growth phase of the virus at 24 h p.i. to circumvent interfering effects like cytopathogenicity and apoptosis (Fig. 4).

Elevated putrescine and N-acetylputrescine levels suggested that SARS-CoV-2 reduces autophagy-inducing polyamine levels (see Fig. 1). Indeed, exogenously applied spermidine and spermine (both 100 µM) inhibited production of infectious SARS-CoV-2 particles by 87% and 54% or GE/ml by 65% and 80%, respectively (Fig. 4, top row, Supplementary Fig. 11a) without causing bovine serum- or ROS-induced cytotoxicity[41] (Supplementary Figs. 10 and 11c). AICAR (25 µM), an AMP analog known to induce autophagy via AMPK activation[42] and inhibit autophagy through promotion of BECN1-VPS34 dissociation[43] had no effect on infectious particle production, but slightly enhanced GE production (Fig. 4, top, Supplementary Fig. 11a). mTORC1 inhibition by rapamycin (0.3 µM), known to induce autophagy, slightly reduced PFU/ml at 24 h p.i. (29%), but increased GE/ml by 101% (Fig. 4, top,

Supplementary Fig. 11a) suggesting that autophagy initiation through inhibition of mTORC1 affects infectious particle production more than virus replication. Limited efficiency of rapamycin might also be connected to saturated amino acid levels (see Fig. 1c). In contrast, reduction of autophagy initiation and nucleation by ULK1 and VPS34 inhibitors (5 μM MRT68921 and 1 μM SAR405, respectively)[44,45] promoted SARS-CoV-2 propagation by up to 492% for MRT68921 (132% GE/ml) and 213% (144% GE/ml) for SAR405. The enhanced infectious particle formation compared to GE/ml suggests that autophagy inhibition plays an important role in virus particle production (Fig. 4, middle, Supplementary Fig. 11a). Interestingly, ULK1 and VPS34 are both involved in the generation of phagophores from ER-derived omegasomes[46] that might compete for lipids, necessary for the generation of convoluted membranes, DMVs, and virus particles. Knockdown of autophagy-initiating ATG5, ATG7, and FIP200, which are responsible for expansion and closure of phagophores, promoted SARS-CoV-2 GE/ml by 488% (ATG5), 354% (ATG7), and 110% (FIP200), confirming the importance of limiting autophagy for successful SARS-CoV-2 propagation (Fig. 4, middle, Supplementary Fig. 11b). Next, we applied the highly selective autophagy-inducing AKT1 inhibitor MK-2206[47] (1 μM), which is currently being tested in a clinical phase II study against breast cancer[48]. AKT1 inhibition results in BECN1 up-regulation and autophagy induction[21,49]. MK-2206 reduced SARS-CoV-2 propagation by up to 92% based on infectious particle formation (46%, GE/ml) further supporting the importance of BECN1 stabilization for CoV inhibition (Fig. 4, bottom, Supplementary Fig. 11a). As expected, knockdown of BECN1 enhanced SARS-CoV-2 replication by 200% (Fig. 4, bottom, Supplementary Fig. 11b). Direct blocking of the negative BECN1 regulator SKP2 by previously described inhibitors SMIP004, valinomycin, and niclosamide[21] showed SARS-CoV-2 growth inhibition (PFU/ml) from 91% (40%, GE/ml) for SMIP004 to >99% (95% and >99.9%, GE/ml) in case of valinomycin and niclosamide (Fig. 4, bottom, Supplementary Fig. 11a). As the highly effective niclosamide is also a vATPase inhibitor and a proton ionophore, acting as oxidative phosphorylation uncoupler[50] that might affect virus entry[51], egress[52], and other stages of the virus life cycle, we further confirmed that niclosamide induces autophagy. Adding BafA1 after niclosamide treatment showed an enhancing effect on the lipidation of LC3B as reflected by comparable LC3B-II/actin ratios between mock- and SARS-CoV-2-infected cells (Supplementary Fig. 11d). Furthermore, we found that a 24 h pre-treatment and complete removal of spermidine, spermine, MK-2206, and niclosamide post SARS-CoV-2 inoculation still showed inhibitory effects, predominantly on virus particle production, suggesting prolonged effects on autophagy-induced virus inhibition (Supplementary Fig. 11e). As VeroFM cells are type I IFN-deficient and have limited expression of the entry co-factor TMPRSS2[29,53], we further confirmed selected findings in human lung Calu-3 cells (Supplementary Fig. 11f). While post-infection treatment with spermine (100 μM) showed SARS-CoV-2 inhibition, spermidine (100 μM) had little or no effect at 24 h p.i. on infectious particle production in Calu-3 cells and a limited inhibitory effect on the virus genome level. However, polyamine levels are cell-type specific and are often elevated in cancer cells (see also Fig. 1)[54,55]. Next, we therefore depleted polyamines by pretreating Calu-3 and VeroFM cells with difluoromethylornithine (DFMO) for 96 h pre-infection (Supplementary Fig. 11g). We confirmed the inhibitory potential of spermidine and spermine in polyamine-depleted Calu-3 and VeroFM cells both showing comparable SARS-CoV-2 inhibition between 84% PFU/ml (10 μM) and 99% (100 μM) for spermidine as well as >97% PFU/ml for both 10 μM and 100 μM spermine (Supplementary Fig. 11g). In agreement with the VeroFM data, post-infection treatment with MK-2206 reduced SARS-CoV-2 in Calu-3 cells by 50% (35% GE/ml). Additionally,

BECN1-stabilizing compounds SMIP004, valinomycin, and niclosamide reduced SARS-CoV-2 by 43% (but increased GE/ml by 23%), 98% (97% GE/ml), and 99% (>99% GE/ml) in Calu-3 cells, respectively (Supplementary Fig. 11f).

**Polyamine supplementation and autophagy induction modulate cellular metabolism and limit SARS-CoV-2 growth in primary human lung cells and intestinal organoids.** To further investigate aspects of drug-induced SARS-CoV-2 inhibition on cellular metabolism, we addressed the polyamine pathway and the AKT1/BECN1 axis by infecting VeroFM and Calu-3 cells with SARS-CoV-2 under treatment with 100 μM spermine (Supplementary Figs. 1–2, PCA, and Supplementary Fig. 12, heatmaps) or 10 μM niclosamide (Supplementary Figs. 1–2, PCA, and Supplementary Fig. 13, heatmaps) and subsequently performed metabolomics 24 h p.i. (Fig. 5a). VeroFM cells showed an increase of most metabolites as compared to untreated SARS-CoV-2-infected VeroFM cells (Fig. 5a, Supplementary Fig. 12) likely by TFEB-dependent upregulation of autophagy genes and re-establishing autophagic flux[15] and metabolite production. In Calu-3 cells, spermine treatment had minor effects on cellular metabolism, which is in agreement with the above-suggested high polyamine baseline levels and limited antiviral effects of exogenous administration. In contrast, niclosamide reverted the SARS-CoV-2-induced increase of metabolites in VeroFM cells (Fig. 5a, Supplementary Fig. 13, upper panels), while maintaining comparably higher metabolite levels in Calu-3 cells (Supplementary Fig. 13, lower panels). In both cell lines, niclosamide treatment resulted in elevated AMP/ATP ratios and reduced levels of TCA cycle-related metabolites (Supplementary Fig. 13), probably by induction of mitochondrial fragmentation[56], mitophagy[57], and stabilization of SKP2/BECN1-dependent autophagy[21].

To confirm that stabilizing polyamine metabolism by exogenous spermine supplementation and inducing autophagy by niclosamide treatment have antiviral effects in complex cell systems, we infected and treated primary human airway epithelial cells (AEC) and intestinal organoids. For both compounds, SARS-CoV-2 inhibition reached 94-99% PFU/ml and >94% GE/ml in AEC at 24 and 48 h p.i. (Fig. 5b, Supplementary Fig. 14a) and 68-86% PFU/ml and 68-94% GE/ml in intestinal organoids at 48 and 72 h p.i. (Fig. 5c, Supplementary Fig. 14b). Finally, for clinical applications, half-maximal inhibitory concentration (IC$_{50}$) of compounds should be in a non-toxic range and reach adequate plasma levels. For the most promising polyamine metabolism- and autophagy-stabilizing compounds, the IC$_{50}$ was 136.7 μM for spermidine, 7.67 μM for spermine, 0.11 μM for MK-2206, and 0.13 μM for niclosamide, based on virus infectious units (Fig. 5d, Supplementary Fig. 14c). Maximal inhibition of infectious virus was achieved with non-toxic concentrations of 333.3 μM for spermidine (85%), 37 μM for spermine (>99%), 3.7 μM for MK-2206 (88%), and 1.24 μM for niclosamide (>99%).

## Discussion

Our data show that highly pathogenic SARS-CoV-2 reprograms the metabolism of cells, induces ubiquitination, and limits autophagy (graphical abstract and working model, Supplementary Fig. 15a, b). SARS-CoV-2 inhibits autophagy as part of its manipulation of the host cellular machinery, whereas the induction of autophagy initiation limits SARS-CoV-2 propagation. Our mechanistic approach, including metabolomics, autophagy signaling, and targeted SARS-CoV-2 inhibition assays, identified two main autophagy-related putative drug targets: the polyamine pathway, and the control of BECN1 abundance through AKT1/SKP2 signaling.

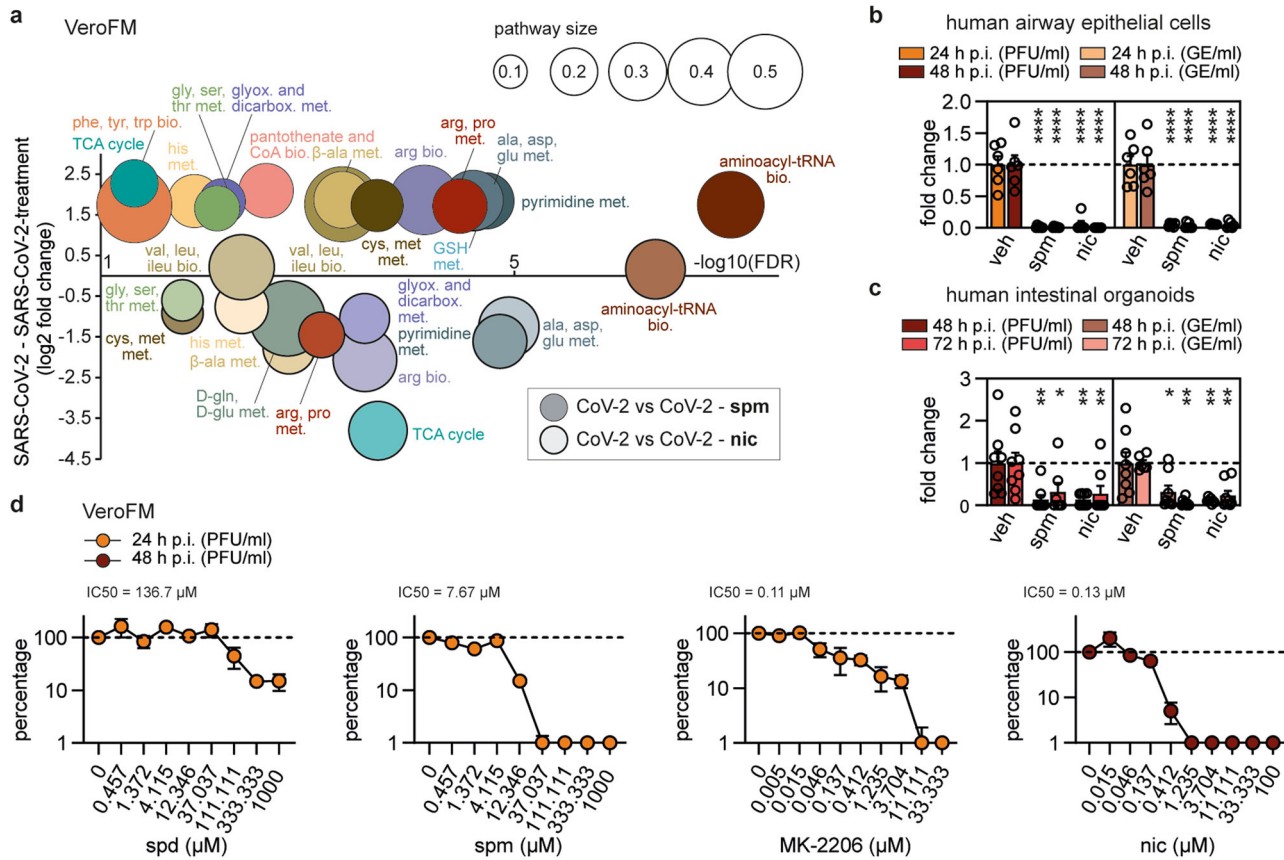

**Fig. 5 Polyamine supplementation and autophagy induction modulate cellular metabolism and limit SARS-CoV-2 growth in primary human lung cells and intestinal organoids. a** Analysis and regulation of significantly altered pathways of SARS-CoV-2-infected VeroFM cells (24 h p.i.) upon treatment with spermine (spm, 100 μM, thin outlined circles) or niclosamide (nic, 10 μM, solid outlined circles). Analysis was done as described in Fig. 1 ($n = 4$ biological independent replicates per group, one experiment). All p-values were determined by a two-way ANOVA and Tukey´s post hoc test. FDRs were adjusted using the Benjamini-Hochberg method. **b** Primary human airway epithelial cells ($n = 6$, from two independent experiments with $n = 3$) were infected with SARS-CoV-2 (MOI = 0.1) and treated with spm, nic, or vehicle as mentioned in **a** for 24 h and 48 h, respectively. P-values were determined by two-way ANOVA, Dunnett's, mean with SEM. **c** Human intestinal organoids ($n = 8$, from two independent experiments with $n = 4$) were infected with SARS-CoV-2 (MOI = 0.05) and treated with spm, nic, or vehicle as above for 48 and 72 h, respectively. Data are presented as fold change PFU/ml or GE/ml. Two-way ANOVA, Dunnett's, mean with SEM was used for p-value calculation. Technical outliers were removed resulting in $n = 7$ for spm, 72 h p.i. (PFU/ml), and nic, 48 h p.i. (GE/ml) as well as $n = 6$ for veh, 72 h p.i. (GE/ml). **d** VeroFM cells were infected with SARS-CoV-2 (MOI = 0.0005) and treated with increasing concentrations of spermidine (spd), spm, MK-2206, and nic ($n = 3$ biological replicates per concentration from one experiment, mean with SEM) to calculate the 50% inhibitory concentration ($IC_{50}$). Technical outliers were removed resulting in $n = 2$ for nic, 0.046 μM, 48 h p.i. (PFU/ml). Limit of detection (LOD) of the plaque assay was 50. Data are presented as percentages of the negative control (see Supplementary Fig. 14 (raw data) and Fig. 10 (cytotoxicity assays)). $p \leq 0.05$ (*), $p \leq 0.01$ (**), $p \leq 0.001$ (***), $p \leq 0.0001$ (****), $p > 0.05$ (not significant, ns). Abbreviations: GE, genome equivalents; LOD, limit of detection; met., metabolism; nic, niclosamide; PFU, plaque-forming units; spd, spermidine; spm, spermine; veh, vehicle.

Regarding the polyamine pathway, we found in vitro and in vivo evidence for transcriptional activation of *SAT1*, which is responsible for acetylation of spermidine and spermine and their conversion to putrescine. Indeed, the putrescine levels were increased in SARS-CoV-2-infected VeroFM and Calu-3 cells suggesting enhanced SAT1 activity. As an ISG, SAT1 showed an overall expression pattern comparable to other ISGs in our in vitro and in vivo data except for ciliated lung cells from mucosal brushes of COVID-19 patients (Fig. 3d). Patients with low SARS-CoV-2 RNA levels expressed significantly more *SAT1* that would reduce spermidine levels and autophagy similar to what we observed in our in vitro experiments. The expression pattern inverted to increased ISG levels but reduced SAT1 in patients with high SARS-CoV-2 RNA concentrations. The virus concentration-dependent transcriptional dynamics suggest direct intervention of SARS-CoV-2 into immune and autophagy gene regulations. High SARS-CoV-2 RNA level-dependent *SAT1* downregulation might be explained by a SARS-CoV-2 papain-like protease-driven degradation of P53[58],

which is key to SAT1 activation[59]. Interestingly, it has previously been shown that spermidine-dependent eIF5A hypusination, which inversely correlates with aging, is also a key factor in efficient B cell activation[15]. Our findings suggest that SAT1 manipulation is a possible mechanism by which SARS-CoV-2 evades a durable antibody response, as preliminary reports have indicated[60]. Additionally, the proposed SARS-CoV and SARS-CoV-2 NSP2 interactions with mitochondrial-residing proteins such as prohibitin (PHB)1 and 2, STOML2, and VDAC2 might limit mitophagy[61–63] and P53-dependent SAT1 activation[64]. Damaged mitochondria might accumulate in the host cell, with consequences for ATP production and release of reactive oxygen species.

Other explanations for limited autophagy in SARS-CoV-2-infected cells include recently described ORF3a, ORF7a, NSP15 involvement[65], and particularly NSP1-dependent host protein translational shutoff[26,27] that might prevent the synthesis of autophagy-regulating proteins, maintain a high ATP/AMP ratio, and increase amino acid availability. The reduction of multiple

autophagy-regulating proteins, including key regulator BECN1, led to reduced ATG14 oligomerization and limited fusion of AP with lysosomes in agreement with previously described mechanisms[22] and similar to MERS-CoV[21]. Noteworthy, despite the relatively global reduction of several autophagy-regulating proteins the most pronounced effect seems to be on the inhibition of AL fusion resulting in P62 and LC3B accumulation in vitro and in vivo that might be explained by a recently discovered ORF3a-mediated mechanism affecting VPS39 and HOPS-RAB7-interaction[66]. The recently described SARS-CoV-2-induced deacidification of lysosomes, necessary for viral egress[52], would further contribute to reduced AL formation, disruption of autophagic flux, accumulation of P62 and LC3B, and subsequently reduced protein degradation. As virus production relies on available amino acids provided either by anabolic synthesis or catabolic recycling processes, the observed global K48-dependent ubiquitination and subsequent proteasomal degradation of cellular proteins might additionally compensate for the limited autophagy-dependent untargeted protein degradation. As global K48-ubiquitination modulates cellular pathways including antiviral IFN and proinflammatory responses[67], deubiquitinating and deISGylating functions of the CoV nsp3-encoded papain-like protease[68] might balance UPS-linked amino acid supply and cellular antiviral countermeasures. In summary, SARS-CoV-2 limits autophagy-dependent protein degradation, possibly to prevent virus particle destruction, drive deacidified lysosome-dependent virus egress[52] and exploit autophagy-related lipid resources for omegasome-dependent DMV formation and virus particle production[69].

The IC$_{50}$ values of the tested compounds indicated efficient virus inhibition in vitro and in ex vivo lung and intestinal models, encouraging clinical trials. Spermidine/spermine plasma levels in humans range between 12–29 ng/ml or 70-280 nM[70,71] being up to 1,000-fold below the detected IC$_{50}$. Such a high IC$_{50}$ might be expected, as spermidine is an endogenous metabolite with intracellular concentrations of 500 μM or more[72]. As long-term spermidine supplementation increased intracellular levels of polyamines in polyamine-deprived cells in old-aged individuals[73], further well-designed in vitro and in vivo studies should clarify putative benefits of polyamine supplementation before or during SARS-CoV-2 infection. Regarding MK-2206, a clinical phase II study found peak plasma concentrations of 0.176 μg/ml (430 nM)[74], which was well above the IC$_{50}$ value of 0.11 μM determined here. A phase I clinical study of niclosamide has reported peak plasma concentrations ranging from 35.7–182 ng/ml (109–556 nM)[75], which encompasses the IC$_{50}$ value of 0.13 μM reported here. Niclosamide is known to have limited resorption efficacy, which might be enhanced by improving the niclosamide formulation or by developing an inhalable derivate for local application[76]. Both MK-2206 and niclosamide might be promising candidates for clinical trials. This work paves the way for the identification and rational design of effective autophagy-modulating antiviral compounds.

## Methods

**Ethics.** The use of primary lung tissue was approved by the Charité ethics committee (projects EA2/079/13 to Andreas Hocke) and written informed consent was obtained from all patients. The use of COVID-19 patient samples applied in histopathology and sequencing was approved by the Ethics Committee of the Charité (EA1/144/13, EA2/066/20 and EA1/075/19 to Helena Radbruch) as well as by the Charité-BIH COVID-19 research board and complied with the Declaration of Helsinki. The generation and cultivation of human primary intestinal organoid cultures was done under the ethics approval no. EA4/164/19 (to Markus Morkel). All animal work was done in compliance with relevant national and international guidelines for care and humane use of animals. The animal use protocol was approved by the Landesamt für Gesundheit und Soziales in Berlin, Germany (approval number 0086/20; approved on 30.04.2020) as described[37].

**Chemicals.** The following chemicals were used for treatment of cells: AICAR (Sigma–Aldrich, A9978), BafA1 (Alfa Aeser, J61835), rapamycin (RAP, Tocris, 1292), niclosamide (NIC, Sigma–Aldrich, Munich, Germany, N3510), valinomycin (VAL, Sigma–Aldrich, V0627), MG132 (Sigma–Aldrich, M8699), MK-2206 HCl (Cayman Chemical, 11593), MRT68921 (Sigma–Aldrich, SML 1644), SMIP004 (Maybridge Chemicals, SPB02305), DSS (disuccinimidyl suberate, Thermo Fisher Scientific, 21655), SAR405 (Sigma–Aldrich, 5.33063), Spermine (Sigma–Aldrich, S4264-5G), Spermidine (Sigma–Aldrich, 85558-5 G), DFMO (Sigma-Aldrich, D193-25MG).

**Cells.** VeroFM (ATCC CCL-81), VeroE6 (ATCC CRL-1586), Calu-3 (ATCC HTB-55) and NCI-H1299 (ATCC CRL-5803) were cultivated in Dulbecco's Modified Eagle´s Medium (DMEM) supplemented with 10% fetal bovine serum (FBS), 1% penicillin/streptomycin, 1% non-essential amino acids, and 1% sodium pyruvate at 37 °C and 5% CO$_2$. To exclude the possibility of FBS-dependent ROS production, pilot experiments with spd and spm were additionally carried out using human serum or aminoguanidine instead of FBS.

**Virus strains and infection.** The applied SARS-CoV-2 strain Munich/2020/984 (BetaCoV/Munich/BavPat1/2020) was isolated from a respiratory swab obtained from the early 2020 Munich patient cohort (GenBank: MT270101; GISAID: EPI_ISL_406862). For virus infection with SARS-CoV-2 strain Munich 984, 2 x 10e5 cells ml-1 were seeded in 6-well plates. After 24 h, cells were infected with an MOI = 0.0005 or 0.1 for infection of primary airway epithelial cells in serum-free medium. In parallel, mock-infected cells were inoculated with heat-inactivated (95 °C, 10 min) virus. After 1 h, the virus dilutions were removed, and the wells were washed twice with PBS and refilled with DMEM (supplemented as described above). Samples were taken at the indicated time points. All virus infection experiments were conducted under biosafety level 3 conditions with enhanced respiratory personal protection equipment. For viral RNA extraction, 50 μl of cell culture supernatant was mixed with 300 μl MagNA Pure external lysis buffer (Roche, Penzberg, Germany) followed by 70 °C for 10 min. Extraction was done by automated pipetting using MagNA Pure 96 instrument (Roche).

**Primary airway epithelial cells and intestinal organoids.** Lung explants were obtained from patients suffering from lung carcinoma, who underwent lung resection at local thoracic surgeries. The study was approved by the Charité ethics committee (projects EA2/079/13) and written informed consent was obtained from all patients. Cells were isolated from tumor-free peripheral lung tissue and cultured as described[77]. For expansion primary cells were co-cultivated with gamma-irradiated mitotically inactivated NIH3T3 mouse embryonic fibroblasts (MEFs) with additional supplementation of 0.1 μM DBZ (Tocris). Differentiation was induced as described and two days prior to infection primary cells were separated from MEF cells by differential trypsinization and seeded in cell culture vessels in DMEM with 10% FCS, 1% non-essential amino acids, 1%, L-glutamine, and 1% sodium pyruvate. Airway epithelial cells were infected with an MOI = 0.1 as described above. Human normal colon organoids were established and cultured as previously published[78]. Normal colon organoids were established from non-cancerous parts of colorectal cancer resection tissue (descending colon), and cultured as previously published. Patient-derived organoid lines were established under ethic´s vote EA4/164/19 of the ethics commission of Charité Universitätsmedizin Berlin. Organoids were harvested, Matrigel (Corning, #356231) was removed by centrifugation and subsequently infected in solution with an MOI = 0.05 as described above. Infected organoids were seeded in Matrigel and medium was supplemented with compounds or vehicle control.

**Real-time reverse-transcription PCR.** SARS-CoV-2 GE/ml were detected by real-time RT-PCR assay targeting SARS-CoV-2 E gene as reported before and listed in Supplementary Table 11[79]. Absolute quantification was performed using SARS-CoV-2-specific in vitro-transcribed RNA standards. Cellular SQSTM RNA (encoding P62) was quantified by real-time RT-PCR using the Superscript III OneStep RT-PCR kit and oligonucleotides p62_f, p62_r, p62_p (listed in Supplementary Table 11). Total RNA was isolated from SARS-CoV-2-infected VeroFM or Calu-3 cells by automated pipetting using the MagNA Pure 96 instrument. SQSTM1 expression was calculated relative to TBP reference gene expression using the ΔΔC$_T$ method[80].

**Plaque assay.** Infectious SARS-CoV-2 plaque-forming units (PFU) were quantified by plaque titration on VeroE6 cells[81]. VeroE6 monolayers were seeded in 24-well plates to reach 90% confluency after 24 h, washed with PBS, incubated with serial dilutions of SARS-CoV-2-containing cell culture supernatants (duplicates, each 200 μl in OPTIPro medium), and overlaid with 1.2% Avicel in DMEM. After 72 h, cells were fixated with 6% formalin and visualized by crystal violet staining. The assay-specific cut-off is 50 PFU/ml.

**siRNA knockdown experiments.** Knockdown experiments were done using dsiRNA transfection[82]. Briefly, 5x10e4 VeroFM cells were seeded in 24-well plates

and transfected using 60 nM siRNA per well using X-tremeGENE™ (Roche) and a reagent/RNA ratio of 3:1, following the manufacturer's instructions. After 48 h the transfection mix was removed, cells were infected with SARS-CoV-2 at an MOI of 0.0005, and samples were taken at 24 h p.i. The following dsiRNAs were obtained from Integrated DNA Technologies (IDT, Leuven, Belgium): ATG5 (CD.Ri.222020.13.1), ATG7 (CD.Ri.222051.13.1), BECN1 (CD.Ri.222059.13.1), FIP200 (CD.Ri.222075.13.3) as well as non-targeting negative control dsiRNAs. Successful knockdown of the indicated targets was verified by Western blot analysis. Protein samples were taken 48 h after siRNA transfection and processed as mentioned below.

**Experimental growth conditions for metabolic profiling and isotope tracing experiments.** In the differential experimental conditions VeroFM and Calu-3 cells were seeded ($5 \times 10^6$ cell/ml) in quadruplicates in 6-well MTPs in the presence of DMEM, DMEM containing 10 μM niclosamide in DMSO, DMEM containing the corresponding amount of DMSO as compared to the 10 μM niclosamide treatment or DMEM containing 100 μM spermine. All these conditions were additionally incubated for 24 h in the presence of SARS-CoV-2 or heat-inactivated SARS-CoV-2 (mock) at an MOI of 0.1. For isotope tracing experiments, Calu-3 cells were seeded ($5 \times 10^6$ cell/ml) in six replicates in 6-well MTPs containing 25 mM U-$^{13}C_6$-labeled glucose (Sigma Aldrich, 660663). The cells were grown for 4 h in the presence of MOI of 0.1 of either SARS-CoV-2 or mock control.

**Metabolite extraction for Liquid Chromatography mass spectrometry (LC-MS).** At the end of each treatment, the growth media from each well was aspirated and the adherent cells were washed twice with pre-warmed (37 °C) wash buffer (75 mM ammonium carbonate (pH 7.4) (VWR NORMAPUR, 21217.260) in Optima LC/MS Grade water, (Thermo Fisher Scientific)). Metabolite extraction from each well was performed for 20 min in 400 μl of a pre-chilled mixture of acetonitrile:methanol:water (40:40:20 [v:v:v] (all solvents Optima LC/MS grade, Thermo Fisher Scientific)) at −20 °C. After removing the extraction solvent and storing it on ice, the extraction was repeated a second time but now the cells were scraped from the plate surface and pooled with the first extract. The combined were subsequently heated to 70 °C for 10 min, to inactivate the virus, before centrifuging them for 10 min at 21,100 g. The metabolite-containing supernatant was transferred to a fresh tube and dried immediately in a Speed Vac concentrator (Eppendorf) before storing it until analysis at −80 °C.

**LC-MS analysis of amine-containing metabolites.** For the analysis of amine-containing compounds, a benzoyl chloride derivatization method[83] was used. Briefly, the dried metabolite pellets were re-suspended in 1500 μl of water (Optima LC/MS grade, Thermo Fisher Scientific). 50 μl of the re-suspended sample was mixed with 25 μl of 100 mM sodium carbonate (Sigma-Aldrich, S7795) in Optima grade LC-MS water (Thermo Fisher Scientific) followed by the addition of 25 μl 2% [v:v] benzoyl chloride (Sigma-Aldrich, B12695) in acetonitrile (Optima LC/MS Grade, Thermo Fisher-Scientific). Samples were vortexed briefly and centrifuged for 10 min at 21,300 g at 20 °C. Clear supernatants were transferred to fresh auto-sampler tubes with conical glass inserts (Chromatographie Zubehoer Trott) and analyzed using an Acquity i-class UPLC (Waters) connected to an Orbitrap ID-X tribrid mass spectrometer (Thermo).

For the analysis 1 μl of the derivatized sample was injected onto a 100 ×2.1 mm HSS T3 UPLC column (Waters). The flow rate was set to 400 μL/min using a buffer system consisting of buffer A (10 mM ammonium formate (Sigma–Aldrich, 70221), 0.15% formic acid (Sigma-Aldrich, 33015) in Optima LC/ MS-grade water, Thermo Fisher Scientific), and buffer B (acetonitrile, Optima LC/MS grade, Thermo Fisher Scientific). The LC gradient was: 0% B at 0 min; 0–15% B 0–0.1 min; 15–17% B 0.1–0.5 min; 17–55% B 0.5–7 min, 55–70% B 7–7.5 min; 70–100% B 7.5–9 min; 100% B 9–10 min; 100-0% B 10–10.1 min, 10.1–15 min 0% B. The mass spectrometer (Orbitrap ID-X, Thermo Fisher Scientific) was operating in positive ionization mode monitoring the mass range m/z 50–750. The heated ESI source settings of the mass spectrometer were: Spray voltage 3.5 kV, capillary temperature 300 °C, sheath gas flow 60 AU, and aux gas flow 20 AU at a temperature of 250 °C. Data analysis was performed using the TraceFinder software (Version 4.1, Thermo Fisher Scientific). Identity of each compound was validated by authentic reference compounds, which were analyzed independently. Peak areas were analyzed by using extracted ion chromatograms (XIC) of compound-specific $[M + nBz + H]^+$ where n corresponds to the number of amine moieties that can be derivatized with benzoyl chloride (Bz). XIC peaks were extracted with a mass accuracy (<5 ppm) and a retention time (RT) tolerance of 0.05 min.

**Anion-Exchange Chromatography-Mass Spectrometry (AEX-MS) of the analysis of TCA cycle and glycolysis metabolites.** Anion-Exchange Chromatography was performed in parallel to the LC-MS analysis of amines. For this purpose, 100 μl of the initially re-suspended sample were analyzed using a Dionex ion chromatography system (Integrion, Thermo Scientific). The applied protocol was adopted from[84]. Briefly, 5 μl of the aqueous metabolite extract were injected in full loop mode using an overfill factor of 3, onto a Dionex IonPac AS11-HC column (2 mm × 250 mm, 4 μm particle size, Thermo Scientific) equipped with a

Dionex IonPac AG11-HC guard column (2 mm × 50 mm, 4 μm, Thermo Scientific). The column temperature was held at 30 °C, while the auto-sampler was set to 6 °C. A potassium hydroxide gradient was generated by the eluent generator using a potassium hydroxide cartridge that was supplied with deionized water. The metabolite separation was carried at a flow rate of 380 μl/min, applying the following gradient. 0–3 min, 10 mM KOH; 3-12 min, 10−50 mM KOH; 21–19 min, 50–100 mM KOH, 19–21 min, 100 mM KOH, 21–22 min, 100–10 mM KOH. The column was re-equilibrated at 10 mM for 8 min. The eluting metabolites were detected in negative ion mode using a Q-Exactive HF mass spectrometer (Thermo Fisher Scientific). The mass spectrometer was operating in negative ionization mode monitoring the mass range m/z 50-750. The heated ESI source settings of the mass spectrometer were: Spray voltage -3.0 kV, capillary temperature 300 °C, sheath gas flow 60 AU, and aux gas flow 20 AU at a temperature of 300 °C.

Data analysis was performed using the TraceFinder software (Version 4.1). Identity of each compound was validated by authentic reference compounds, which were analyzed independently. Peak areas were analyzed by using extracted ion chromatograms (XIC) of compound-specific $[M – H^+]^-$. XIC peaks were extracted with a mass accuracy (<5 ppm) and a retention time (RT) tolerance of 0.1 min.

**$^{13}C$ Isotope enrichment analysis after U-$^{13}C_6$ Glucose tracing.** $^{13}C$ isotope enrichments for all detectable isotopologues were performed using the TraceFinder (Version 4.1). Identity of each compound was validated by reference compounds using either the $[M–H^+]^-$ ion from anion chromatographic data or the $[M + nBz+H^+]^+$ ion from the data obtained from the benzoyl-derivatized amines analysis.

For the differential isotope enrichment analysis the sum of all peak areas of the extracted isotopes of a compound was calculated and the fraction of the individual isotope to the sum of isotopes was calculated (isotope enrichment analysis). Based on the relative enrichment of each isotopologue the fraction of the labeled isotopes of each compound was determined. This atom fraction labeled represent the relative contribution of stable $^{13}C$ isotopes to the total area of each compound (denoted as atom fraction labeled in Supplementary Fig. 4a-b). Isotopologue peaks were extracted using a mass accuracy (<5 ppm) and a retention time tolerance of <0.05 min.

**Analysis of metabolomics data.** Identification of significant metabolite level alterations: Metabolite intensities were log2 transformed and Pareto-scaled for further statistical analysis. Significant metabolite level changes upon 24 h p.i. compared to mock were identified by Student's T-test with an adjusted p-value (false discovery rate (FDR)) ≤0.05 using MetaboAnalyst[85]. Identification of significantly overrepresented pathways: Metabolomics pathway analyses were performed with the significantly altered metabolites from Supplementary Tables 1 and 2 using MetaboAnalyst applying a Fisher's Exact Test and Out-degree Centrality for pathway topology analysis. Pathways were considered for further analyses with an FDR ≤ 0.1[85]. Heat map data (Supplementary Figs. 12 and 13) was generated from log10 transformed and z-score scaled data using the InstantClue v.0.10.10.dev-snap software[86].

**Toxicity assays.** Presto Blue cell viability and LDH cytotoxicity assays in VeroFM or Calu-3 cells were carried out according to the manufacturer's protocols provided by ThermoFisher Scientific and Takarabio, respectively. Briefly, for PrestoBlue cell viability assay, cells were incubated in the presence of 10 μl PrestoBlue (10x) reagent for 30 min at 37 °C and 5% CO₂ after the indicated treatment times. 570 nm absorbance values were normalized to 600 nm values for further analysis. For LDH cytotoxicity assays, 100 μl of cell-free supernatant was measured for LDH activity at 492 nm after the indicated treatment times. A wavelength >600 nm served as reference wavelength. For MTT tests, cells were incubated in the presence of 0.5 mg ml$^{-1}$ tetrazole 3-(4,5-dimethylthiazol-2-yl-)-2,5-diphenyltetrazolium bromide (MTT) for 4 h at 37 °C and 5% CO₂. Read-out of MTT assay was carried out as described in the manual instructions[21].

**Protein analysis, Western blot and Atg14 oligomerization.** Protein extracts were obtained by lysing cells in Pierce IP Lysis buffer freshly supplemented with protease inhibitor cocktail 3 (Merck Millipore, Darmstadt, Germany), and phosphatase inhibitor PhosSTOP (Roche, Penzberg, Germany). Subsequently, SDS loading buffer (4x NuPAGE LDS, ThermoFisher Scientific) was added and samples were heated to 95 °C for 10 min to inactivate SARS-CoV-2. Proteins were separated by SDS-PAGE and electro-transferred onto PVDF membranes[21]. Blots were placed in Tris-buffered saline, supplemented with 0.05% Tween (Sigma Aldrich) and 5% non-fat milk for 1 h at room temperature, and then incubated with primary antibody (diluted in TBS/0.05% Tween) overnight at 4 °C. Subsequently, blots were washed and probed with the respective horseradish peroxidase- or fluorophore-conjugated secondary antibody for 1 h at room temperature. The immuno-reactive bands were visualized either using ECL detection reagent (Millipore, Billerica, MA, USA) or directly by excitation of the respective fluorophore. Determinations of the band intensities were performed with BioRad, ChemiDoc MP.

The following primary antibodies were used: β-actin (1:5,000 Cell Signaling Technology, #8457), SQSTM1/p62 (1:1,000, Cell Signaling Technology, #5114), LC3B (1:1,000, Cell Signaling Technology, #3868), BECN1 (1:1,000, Cell Signaling

Technology, #3738), pBECN1[S15] (1:1,000, Cell Signaling Technology, #84966), ATG14 (1:1,000, Cell Signaling Technology, #5504), pATG14[S29] (1:1,000, Cell Signaling Technology, #13155), ULK1 (1:1,000, Cell Signaling Technology, #8054), pULK1 (S555) (1:1,000, Cell Signaling Technology, #5869), pULK1 (S757) (1:1,000, Cell Signaling Technology, #6888), TSC2 (1:1,000, Cell Signaling Technology, #3612), pTSC2 (S1387) (1:1,000, Cell Signaling Technology, #5584), AMPKα (1:1,000, Cell Signaling Technology, #2532), pAMPK (T172) (1:1,000, Cell Signaling Technology, #2531), pAMPK substrate motif (1:1,000, Cell Signaling Technology, #5759), pAKT (S473) (1:1,000, Cell Signaling Technology, #4060), AKT (1:1,000, Cell Signaling Technology, #9272), HSC70 (1:5,000, Enzo Life Sciences, ADI-SPA-757-F), SKP2 (1:1,000, Cell Signaling Technology, #2652), pSKP2 (S64) (1:1,000, Cell Signaling Technology, #14865). Anti-pSKP2 (S72) was a kind gift by Cell Signaling Technology (used 1:1,000). Secondary antibodies: anti-rabbit IgG, HRP-linked antibody (1:10,000, Cell Signaling Technology, #7074), anti-mouse IgG, HRP-linked antibody (1:10,000, Cell Signaling Technology, #7076). The immunoreactive bands were visualized using ECL detection reagent (BioRad, Hercules, CA, USA). Determination of the band intensities was performed with BioRad, ChemiDoc MP. In general, protein quantification was performed by normalization to the intensity of actin or HSC70, which was determined on the same membrane. For quantification of phosphorylated proteins, this signal was always referred to the signal intensity of the corresponding total protein. For ATG14 oligomerization, the PBS-washed cell pellet was incubated with 75 µM DSS (disuccinimidyl suberate, Thermo Fisher Scientific, 21655) or corresponding vehicle (DMSO) in PBS. Crosslinking was performed for 30 min at room temperature followed by 2 h, at 4 °C. Crosslinking was quenched in Tris-buffered saline (pH 7.0) for 20 min at 4 °C. Protein extracts were analyzed by capillary electrophoresis on Wes™ (ProteinSimple) using the 60–440 kDa cartridges.

**Autophagic flux, GFP-FYVE, and immunofluorescence test.** To determine the effect of drug treatment or CoV infection on autophagic flux, DMSO (vehicle control) or BafA1 were applied two hours before cell lysis, followed by Western blot analysis as described. In order to detect autophagosomes and autophagolysosomes by immunofluorescence test, VeroFM and NCI-H1299 cells were transfected with ptfLC3 plasmid (Addgene.org, #21074) that expresses LC3 tagged with both GFP (inactivated in autolysosomes) and mRFP (resists inactivation in autolysosomes)[36], following the guidelines for monitoring autophagy[32]. For transfections, cells were treated with 0.5 µg plasmid and 1.5 µl Fugene HD (Promega, Mannheim, Germany) 24 h prior to infection with SARS-CoV-2 (MOI = 0.001) and drug treatment the next day. Cells were fixed (6% formaldehyde for 1 h) at indicated time points (24 h) and analyzed by fluorescence microscopy (AxioVert 200 M, Carl Zeiss, Oberkochen, Germany) equipped with a plan-Apochromat 63x/1.40 Oil DIC objective and an AxioCam MR R3 camera. Images were acquired and processed using AxioVision software ZEN Pro 2 (Carl Zeiss). A scientist blind to the conditions counted vesicles in 40-50 randomly selected cells. For GFP-FYVE detection VeroFM cells were transfected with GFP-FYVE and infected with SARS-CoV-2. After 24 h p.i. cells were fixed and stained with an undiluted hybridoma culture supernatant of an in-house mouse monoclonal anti-SARS-CoV-2 nucleo-capsid antibody (#G229FA10; kindly provided by J. A. Schenk and F. Sellrie; UP Transfer GmbH, Potsdam). Secondary antibody detection was done with Cy3-labeled polyclonal goat-anti mouse antibody (1:200, Dianova, 115-165-166).

**Autopsy lung tissue.** This study was approved by the Ethics Committee of the Charité (EA 1/144/13, EA2/066/20 and EA1/075/19) as well as by the Charité-BIH COVID-19 research board and complied with the Declaration of Helsinki. For histology, we analyzed cryopreserved lung tissue from deceased patients that were tested positive for SARS-CoV-2 RNA via PCR[79]. Lung tissue from deceased patients with clinical and autoptic signs of non-COVID pneumonia (n = 3) and without signs of pneumonia (n = 3) was used as controls. COVID-19 patients were randomly selected out of our autopsy cohort (n = 6 out of 25)[87]. Inclusion criteria were presence of cryopreserved material and viral RNA load in the lungs. Patients were excluded if tumor infiltration could be detected in the samples and one case with graft vs host reaction after stem cell therapy. For single-nucleus (sNuc) RNA sequencing cases were selected using the same inclusion and exclusion criteria, but not randomly selected. Instead, cases were selected with respect to the expected RNA quality and disease duration (early = <14 days, and late = >14 days disease duration, in total n = 7 out of 25). Clinical records were assessed for pre-existing medical conditions and medications, current medical course, and ante-mortem diagnostic findings as specified in Supplementary Table 7. All specimens were cryopreserved after removal at −80 °C prior to diagnostic work-up.

**Morphological Analysis & semi-quantitative assessment of autopsy lung tissue.** All stains were performed on 8µm cryomicrotome sections, with a NX80 cryotome (ThermoFisher, Waltham, Massachusetts, USA) according to standard procedures. Stains included Hematoxylin and eosin (H&E), LC3 (2G6, nanoTools, 1:50) and p62 (SQSTM1, abcam, 1:100). Immunohistochemical stainings were performed on a Benchmark XT autostainer (Ventana Medical Systems, Tuscon, AZ, USA). For each specimen, four random fields of vision were analyzed and photographed at 40x magnification with an Olympus BX50 microscope, the digital camera DP25, and cell D software. Positively stained cells were defined by presence of a nucleus and cytoplasmic staining, counted independently by T.A. and H.R., and summed together for the four fields of vision.

**Single-cell (Sc)/single-nucleus (sNuc) RNA sequencing. ScRNA-seq from olfactory mucosal swabs.** Previously published scRNA-seq data of olfactory mucosal swabs from uninfected and infected (moderate and critical cases) SARS-CoV-2 patients were re-analyzed to assess autophagic factors and their gene expression levels in vivo[38]. After selecting for patient samples that were obtained before 14 days since the onset of symptoms, we analyzed a total of 27,876 cells split between SARS-CoV-2 with low viral load ("low", n = 5, <10e5 SARS-CoV-2 RNA copies per ml), and SARS-CoV-2 with high viral load ("high", n = 3, >10e5 SARS-CoV-2 RNA copies per ml). Detailed patient information can be found in Supplementary Table 8.

**SNuc-seq from postmortem lungs.** For sNuc-seq, fresh frozen lung samples were used from deceased COVID-19 patients with an early time point of death ("early", n = 3) and a late time point of death ("late", n = 4). The patients were stratified according to the duration of the disease with early = <14 days, and late = >14 days. Nuclei were isolated and prepared for sNuc-seq RNA sequencing as previously described[88]. Briefly, 50-60 µm lung tissue sections were homogenized using the loose and tight pestles of glass tissue douncers in citric acid buffer (250 mM sucrose, 25 mM citric acid, 1 µg/mL Hoechst 33342). The homogenate was successively passed through a 100 µm and a 35 µm cell strainer to remove large debris. This was then centrifuged for 5 min at 500 g at 4 °C. The nuclei were then washed in 1 ml of citric acid buffer. To remove high amounts of debris, the nuclei were passed through a density centrifugation gradient by adding an equal volume of S88 citric acid buffer (882 mM sucrose, 25 mM citric acid) to the bottom of the tube before being centrifuged for 10 min at 1,000 g at 4 °C. The supernatant was carefully removed, and the nuclei were resuspended in cold resuspension buffer (25 mM KCl, 3 mM MgCl2, 50 mM Tris-buffer, 0.4 U/µL RNaseIn Takara 2313 A, 1 mM DTT, 0.4 U/µl SuperaseIn Thermo Fisher Scientific AM2694, 1 µg/mL Hoechst 33342). The nuclei were counted and then immediately loaded into the 10x Chromium Controller using the 10x Genomics Single Cell 3' Library Kit v3.1 (10x Genomics; PN 1000223; PN 1000157; PN 1000213; PN 1000122) and the subsequent reverse transcription, cDNA amplification, and library preparation were performed according to the manufacturer's instructions with minor modifications. Specifically, the 85 °C incubation step during the reverse transcription step was extended to 10 min and an additional 2 cycles were added to the cDNA amplification. Afterwards, the 3'RNA sequencing libraries were pooled and sequenced on the NovaSeq 6000 Sequencing System (Illumina, paired-end, single-indexing). Single-nucleus postmortem lung dataset was processed using cell ranger 3.0.1. All transcripts were aligned to a customized hg19 reference transcriptome (10x Genomics, version 3.1.0) that included the SARS-CoV-2 genome (Refseq-ID: NC_045512) as an additional chromosome. Further pre-processing was performed with Seurat 3.1.4[89,90] where cells with more than or equal to 10% mitochondrial reads or less than 200 genes expressed were filtered out. An upper cutoff for the number of genes was selected manually for each sample based on the outliers in a UMI counts vs. gene counts plot (3,000-6,000 genes). After log-normalization, samples were integrated with canonical correlation analysis (CCA) in Seurat. Using the integrated dataset, PCA was run, followed by UMAP calculation and finally clustering. Cell type assignment was performed as previously described[38,91] and by using marker genes[92]. The postmortem lung dataset is depicted in a UMAP with the identified cell types identified in uninfected and infected patients with SARS-CoV-2 (Supplementary Fig. 9a–i). "SARS2high" cells exhibit the expression of AT1, AT2, and secretory cell markers, which may indicate that this cluster of cells is composed of different cell types sharing a similar transcriptome due to infection (Supplementary Fig. 9d). Of note, due to the nature of the sNuc-seq technique, cytoplasmic transcripts are lost which consequently includes viral RNA transcripts. It is possible that some SARS-CoV-2+ cells were co-captured with ambient viral RNA. SARS-CoV-2-positive cells are defined as any cell with >1 viral RNA transcript. Some identified cell types are present in different states of maturation as indicated by differing levels of marker gene expression (Supplementary Fig. 9h). The differences in the QC metrics of the patient samples might be explained by the time point of sampling after death (Supplementary Table 7). After pre-processing and filtering, we analyzed 51,138 nuclei.

**Statistics and Reproducibility.** When two groups were compared, the Student's t-test was applied. For three or more group comparisons, one or two-way analysis of variance (ANOVA) was performed, as appropriate, followed by Tukey's or Bonferroni's or Sidak and Dunnett´s post hoc tests, as appropriate. All t test and p-values and ANOVA F and p-values are reported in the legends to the Figures and Supplementary Figures; significant results of the contrast tests are further indicated by asterisks in the graphs. All statistical tests were two-tailed and $p < 0.05$ was considered statistically significant. The asterisks indicate $p \leq 0.05$ (*), $p \leq 0.01$ (**), $p \leq 0.001$ (***), $p \leq 0.0001$ (****), $p > 0.05$ (not significant, ns). For the complete set of raw data, see the data source file. For PCA analysis log10 transformed and pareto scaled and mean-centered peak areas of all samples were analyzed using SIMCA 13 (Umetrics). Differential gene expression testing was calculated using FindMarkers with the MAST-based differential expression test. The disease

duration (postmortem lung) and days post onset of symptoms (olfactory mucosa) were considered as latent variables. *P*-values were adjusted using the Benjamini-Hochberg method. In all cases when virus plaques were not detectable, we independently calculated the p-values by considering either the limit of detection of 50 or 0 PFU/ml. Only when both calculations had a $p < 0.05$ we included a p-value. In case of the $IC_{50}$ calculations (Fig. 5d), the end-point was defined as the highest compound concentration that showed no plaques in all $n = 3$ samples.

**Reporting summary**. Further information on research design is available in the Nature Research Reporting Summary linked to this article.

## Data availability

Metabolomics data shown in Figs. 1 and 5, Supplementary Figs. 1–4, 12–13, and Supplementary Tables 1–4 are available at the Metabolomics data repository MetaboLights under the study identifier MTBLS2840. Histopathological lung datasets generated and analyzed during the current study (Fig. 3b) are available on reasonable request. The sequencing data of Fig. 3c, and Supplementary Fig. 9 are available under controlled access and require a Data Transfer Agreement in the European Genome-pheome Archive repository: EGAS00001004689. RNAseq data mentioned in Supplementary Figs. 5 and 8 are deposited: GSE148729, GSE147507, and GSE162208. Source data are provided with this paper.

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

## Acknowledgements

We thank Patricia Tscheak and Antje Richter (both Charité) for technical assistance and Terry Jones for editing the manuscript. **Funding:** N.C.G and M.A.M have financial support from BMBF-ProteoCoV 01KI20434A and B, respectively. C.D. is supported by BMBF-RAPID 01KI1723A. C.D., J.T., and N.O. were supported by COVID-19 funds made available through the Berlin University Alliance and Freie Universität Berlin. K.H. was supported by BMBF (Camo-COVID-19) and DFG (SFB-TR 84, A07). C.N. was supported by an Argelander Grant awarded by the University of Bonn. A.C.H. was supported by the Berlin University Alliance GC2 Global Health (Corona Virus Pre-Exploration Project), by BMBF (RAPID, Organo-Strat, alvBarriere-COVID-19), by DFG (SFB-TR84; B6, Z1a), by the Berlin Institute of Health (BIH), Charité 3 R, and Charité-Zeiss MultiDim. Work in the T.F.M. laboratory was supported by the State of Schleswig-Holstein (Covid-19 DIO47). The authors are most grateful to the patients and their relatives for consenting to autopsy and subsequent research, which were facilitated by the Biobank of the Department of Neuropathology – Universitätsmedizin Berlin, Germany.

## Author contributions

N.C.G. and M.A.M. designed and conceived the work. N.C.G., J.P., T.B., J.E., F.D., R.L.C., J.T., N.H., C.N., F.W., K.H., T.A., D.E.H., T.E., A.Z., M.L., E.W., S.S., A.R., M.M., H.R., P.G. and M.A.M. carried out experiments. N.C.G., J.P., T.B., J. E., F.D., R.L.C., J.T., N.H., C.N., F.W., K.H., T.A., D.E.H., K.W., T.E., A.Z., M.L., E.W., S.S., A.R., D.N., K.H., T.F.M., F.H., V.M.C., M.L., A.H., M.M., N.O., C.C., R.E., H.R., P.G., C.D. and M.A.M analyzed data or contributed essential material. N.C.G. and M.A.M. wrote the main paper text. N.C.G. and T.B. prepared all figures. All authors reviewed the paper.

## Funding

## Competing interests

The authors declare no competing interests.

## Additional information

Nils C. Gassen [1✉], Jan Papies[2,3], Thomas Bajaj[1], Jackson Emanuel [2,3], Frederik Dethloff[4], Robert Lorenz Chua [5], Jakob Trimpert [6], Nicolas Heinemann[2,3], Christine Niemeyer[1], Friderike Weege [2,3], Katja Hönzke[7], Tom Aschman [8], Daniel E. Heinz [1], Katja Weckmann[1], Tim Ebert[1], Andreas Zellner[1], Martina Lennarz[1], Emanuel Wyler [9], Simon Schroeder[2,3], Anja Richter[2,3], Daniela Niemeyer [2,3], Karen Hoffmann[7], Thomas F. Meyer[10], Frank L. Heppner [8,11,12], Victor M. Corman [2,3], Markus Landthaler [9,13], Andreas C. Hocke[7], Markus Morkel [14,15], Nikolaus Osterrieder[6,16], Christian Conrad[5], Roland Eils [5,17,18], Helena Radbruch[8], Patrick Giavalisco[4], Christian Drosten [2,3] & Marcel A. Müller [2,3,19✉]

[1]Department of Psychiatry and Psychotherapy, University of Bonn, Medical Faculty, Bonn, Germany. [2]Institute of Virology, Charité-Universitätsmedizin Berlin, corporate member of Freie Universität Berlin, Humboldt-Universität zu Berlin, Berlin, Germany. [3]German Center for Infection Research (DZIF), partner site Charité, Berlin, Germany. [4]Max Planck Institute for Biology of Ageing, Cologne, Germany. [5]Center for Digital Health, Berlin Institute of Health (BIH) and Charité-Universitätsmedizin Berlin, corporate member of Freie Universität Berlin, Humboldt-Universität zu Berlin, Berlin, Germany. [6]Institute of Virology, Freie Universität Berlin, Berlin, Germany. [7]Molecular Imaging of Immunoregulation, Medizinische Klinik m.S. Infektiologie & Pneumologie, Charité-Universitätsmedizin Berlin, Berlin, Germany. [8]Department of Neuropathology, Charité-Universitätsmedizin Berlin, corporate member of Freie Universität Berlin and Humboldt-Universität zu Berlin, Berlin, Germany. [9]Berlin Institute for Medical Systems Biology, Max-Delbrück-Center for Molecular Medicine in the Helmholtz Association, Berlin, Germany. [10]Laboratory of Infection Oncology, Institute of Clinical Molecular Biology, UKSH, Christian Albrechts University of Kiel, Kiel, Germany. [11]German Center for Neurodegenerative Diseases (DZNE) Berlin, Berlin, Germany. [12]Cluster of Excellence, NeuroCure, Berlin, Germany. [13]IRI Life Sciences, Institut für Biologie, Humboldt-Universität zu Berlin, Berlin, Germany. [14]Institute for Pathology, Charité-Universitätsmedizin Berlin, corporate member of Freie Universität Berlin, Humboldt-Universität zu Berlin, Berlin, Germany. [15]German Cancer Consortium (DKTK) Partner Site Berlin, German Cancer Research Center (DKFZ), Heidelberg, Germany. [16]Department of Infectious Diseases and Public Health, Jockey Club College of Veterinary Medicine and Life Sciences, City University of Hong Kong, Kowloon Tong, Hong Kong. [17]German Center for Lung Research (DZL), Berlin, Germany. [18]Data Science Unit, Heidelberg University Hospital and BioQuant, Heidelberg, Germany. [19]Martsinovsky Institute of Medical Parasitology, Tropical and Vector Borne Diseases, Sechenov University, Moscow, Russia. ✉email: nils.gassen@ukbonn.de; marcel.mueller@charite.de

