## [Peer Review File · Nature Communications]

Reviewers' Comments:

Reviewer #1:

Remarks to the Author:

The authors made significant changes to better present metabolic data. Specifically, they changed some wording to reflect that the data depict "isotope enrichment" and not "metabolic fluxes". Further, they also reanalyzed the metabolic tracing data and now present data as barplots (instead of a heatmap). However, the ^{13}C glucose data do not add significantly to the manuscript regarding TCA cycle metabolism as ^{13}C glucose barely labels TCA cycle intermediates in VeroFM cells. It is not clear if glucose oxidation and/or oxidative metabolism are affected in those cells under the condition tested. Based on the current ^{13}C tracing data, glucose oxidation is very different in the two cell lines tested (VeroFM and Calu3), even in mock conditions that may influence some of the metabolic conclusions. Overall, the paper focuses on autophagy, and the broad claims of metabolic changes require more in-depth metabolic studies.

1. Extended Fig. 4a now depicts atom fraction enrichment from ^{13}C glucose. It is of concern that ^{13}C glucose labels TCA cycle intermediates less than 2% (though pyr is 20% labeled) in VeroFM cells. The authors depict aKG, mal, and fum only. No data on citrate is provided. The labeling around 1% is very low and might indicate background labeling. Did the author confirm labeling with non-labeled control samples corrected for natural isotope abundance? In contrast, citrate and other TCA cycle intermediates were labeled up to 25% in Calu-3 cells (Ext Fig 4b).

2. Without proper controls, the current data suggest that VeroFM cells barely oxidize any glucose in mock and infected conditions. As such, conclusions regarding anabolism/catabolism and TCA cycle metabolism are challenging using ^{13}C glucose tracer. Cells may use other substrates to fuel TCA cycle intermediates, such as glutamine. Is respiration affected in those cell models?

Rebuttal letter Nature comm Gassen et al.

Reviewer #1 (Remarks to the Author):

The authors made significant changes to better present metabolic data. Specifically, they changed some wording to reflect that the data depict “isotope enrichment” and not “metabolic fluxes”. Further, they also reanalyzed the metabolic tracing data and now present data as barplots (instead of a heatmap). However, the ¹³C glucose data do not add significantly to the manuscript regarding TCA cycle metabolism as ¹³C glucose barely labels TCA cycle intermediates in VeroFM cells. It is not clear if glucose oxidation and/or oxidative metabolism are affected in those cells under the condition tested. Based on the current ¹³C tracing data, glucose oxidation is very different in the two cell lines tested (VeroFM and Calu3), even in mock conditions that may influence some of the metabolic conclusions. Overall, the paper focuses on autophagy, and the broad claims of metabolic changes require more in-depth metabolic studies.

Reply: *We thank the reviewer for the constructive and helpful comments. We fully agree with the reviewer that the paper focuses on autophagy and that the metabolomics provide an essential asset to the paper’s conclusions, regarding the global increase of amino acid, putrescine, and nucleoside triphosphate levels in SARS-CoV-2-infected VeroFM and Calu-3 cells. We agree with the reviewer that our metabolomics dataset is insufficient to draw overarching conclusions about the nature of host anabolism and catabolism during SARS-CoV-2 infection and have revised our statements accordingly. Furthermore, we agree that the reviewer is also correct in their assessment of the ¹³C labeling in VeroFM cells. We were also surprised by the paucity of labelled TCA cycle intermediates and cannot exclude the possibility of a technical problem that limited ¹³C incorporation or detection. Therefore, as we lack the capacity to experimentally contextualize these findings in a timely manner, we have elected to remove the ¹³C VeroFM data from the revised manuscript.*

1. Extended Fig. 4a now depicts atom fraction enrichment from ¹³C glucose. It is of concern that ¹³C glucose labels TCA cycle intermediates less than 2% (though pyr is 20% labeled) in VeroFM cells. The authors depict aKG, mal, and fum only. No data on citrate is provided. The labeling around 1% is very low and might indicate background labeling. Did the author confirm labeling with non-labeled control samples corrected for natural isotope abundance? In contrast, citrate and other TCA cycle intermediates were labeled up to 25% in Calu-3 cells (Ext Fig 4b).

Reply: *The reviewer’s observation concerning the ¹³C glucose enrichment in mitochondrial metabolites was indeed quite unexpected. To further validate the lack of ¹³C glucose enrichment in mitochondrial metabolites we compared the atom fraction labeled of the ¹³C fed cells to cells fed with ¹²C glucose as control. As predicted by the reviewer, the ¹³C enrichment is nearly identical to background labeling. Therefore, we cannot assert much about the differential effect of SARS-CoV-2 infection on oxidative phosphorylation in the mitochondria of the VeroFM cells in this experiment, but we can clearly see, similar to the results in the Calu3 cells, that glycolysis, nucleotide biosynthesis and amino acid biosynthesis, by and large, are unaffected by SARS-CoV-2 infection. Although the lack of mitochondrial metabolites is an interesting phenomenon, we currently cannot exclude technical issues (e.g. BSL3-related heat-inactivation) and we would have to repeat the complete experiment and the analysis. As the revised version of this paper strongly focuses on autophagy, we would rather choose to take out the VeroFM ¹³C glucose tracing data. In the revised manuscript, we reduced the Calu-3-based glucose tracing data (**line 152**) and deleted all claims comparing anabolism vs catabolism in a general way. The revised manuscript restricts its focus to our autophagy findings.*

The text was changed accordingly:

Line 153: To explore if the increased metabolite concentrations resulted from increased anabolism or catabolism, we performed a 24-hours post infection isotope enrichment analysis using U-¹³C₆ glucose as a metabolic tracer. Interestingly, the isotope tracing experiment showed only small and mostly insignificant differences in the relative isotope enrichment between infected and mock cells (**Extended data Fig. 4a-b; Extended Data Tables 3 (VeroFM) and 4 (Calu-3)**). This result therefore implies, due to the large significantly increased metabolite concentrations in the SARS-CoV-2-infected cells (**Extended data Fig. 3**), that de novo synthesis has a certain impact on these observed metabolic upregulations (**Extended data Fig. 5; Extended Data Tables 5 (VeroFM) and 6 (Calu-3)**). Still, the observed systemic increase in amino acids (**Fig. 1c, orange**) cannot be derived solely from amino acid synthesis and must be supported by amino acid accumulation due to protein degradation by either the UPS and/or autophagy.

Was changed to:

Line 152: To explore if high amino acid levels result from de novo biosynthesis, we performed U-¹³C₆ glucose isotope enrichment analysis in SARS-CoV-2- and mock-infected Calu-3 cells (**Extended Data Fig. 4a-b; Extended Data Tables 3-4**). As relative ¹³C enrichment of the most abundant amino acids and intermediates of glycolysis and the TCA cycle (**Extended Data Fig. 4a-b**) showed minor differences between the SARS-CoV-2- and mock-infected cells, we focused our analyses on catabolic protein degradation via the UPS or autophagy.

2. Without proper controls, the current data suggest that VeroFM cells barely oxidize any glucose in mock and infected conditions. As such, conclusions regarding anabolism/catabolism and TCA cycle metabolism are challenging using ¹³C glucose tracer. Cells may use other substrates to fuel TCA cycle intermediates, such as glutamine. Is respiration affected in those cell models?

Reply: This is indeed a possibility but we believe that expanding the experimental data towards ¹³C Gln tracing would be beyond the scope of the current manuscript. Measuring respiration in SARS-CoV-2-infected cells would require a way to assess cellular respiration in a BSL3 setting. Although we plan to perform such experiments in the near future, we would ask to publish the manuscript in its current revised version.